# The BrowserGym Ecosystem for Web Agent Research

**Thibault Le Sellier De Chezelles**[*123]  **Maxime Gasse**[*123]  **Alexandre Lacoste**[*1]

Core contributors:
**Alexandre Drouin**[12]  **Massimo Caccia**[1]
**Léo Boisvert**[123]  **Megh Thakkar**[12]  **Tom Marty**[27]  **Rim Assouel**[27]  **Sahar Omidi Shayegan**[125]

Benchmark contributors:
**Lawrence Keunho Jang**[4]  **Xing Han Lù**[25]  **Ori Yoran**[6]  **Dehan Kong**[8]  **Frank F. Xu**[4]

Affiliated advisors:
**Siva Reddy**[125]  **Quentin Cappart**[23]  **Graham Neubig**[4]  **Ruslan Salakhutdinov**[4]  **Nicolas Chapados**[123]

Lead:
**Maxime Gasse**[123]  **Alexandre Lacoste**[1]

[1]*ServiceNow Research* [2]*Mila* [3]*Polytechnique Montréal*
[4]*Carnegie Mellon University* [5]*McGill University* [6]*Tel Aviv University* [7]*Université de Montréal* [8]*iMean AI*

Reviewed on OpenReview: `https://openreview.net/forum?id=5298fKGmv3`

## Abstract

The BrowserGym ecosystem addresses the growing need for efficient evaluation and benchmarking of web agents, particularly those leveraging automation and Large Language Models (LLMs) for web interaction tasks. Many existing benchmarks suffer from fragmentation and inconsistent evaluation methodologies, making it challenging to achieve reliable comparisons and reproducible results. In an earlier work, Drouin et al. (2024) introduced BrowserGym which aims to solve this by providing a unified, gym-like environment with well-defined observation and action spaces, facilitating standardized evaluation across diverse benchmarks. We propose an extended BrowserGym-based ecosystem for web agent research, which unifies existing benchmarks from the literature and includes AgentLab, a complementary framework that aids in agent creation, testing, and analysis. Our proposed ecosystem offers flexibility for integrating new benchmarks while ensuring consistent evaluation and comprehensive experiment management. This standardized approach seeks to reduce the time and complexity of developing web agents, supporting more reliable comparisons and facilitating in-depth analysis of agent behaviors, and could result in more adaptable, capable agents, ultimately accelerating innovation in LLM-driven automation. As a supporting evidence, we conduct the first large-scale, multi-benchmark web agent experiment and compare the performance of 6 state-of-the-art LLMs across 6 popular web agent benchmarks made available in BrowserGym. Among other findings, our results highlight a large discrepancy between OpenAI and Anthropic's latests models, with Claude-3.5-Sonnet leading the way on almost all benchmarks, except on vision-related tasks where GPT-4o is superior. Despite these advancements, our results emphasize that building robust and efficient web agents remains a significant challenge, due to the inherent complexity of real-world web environments and the limitations of current models.

## 1 Introduction

In recent years, the emergence of powerful large language models (LLMs) and vision-language models (VLMs) (OpenAI, 2024a; Anthropic, 2024a; Meta, 2024) has led to a revolution in the field of conversational assistants,

---

*Equal contribution.

otherwise known as chatbots. Since then, the capabilities of these traditional, chat-only assistants have been expanded to include running search queries on the web, reading user documents, executing sand-boxed code, and generating images. Of particular interest in this work is the capability for conversational assistants to perform actions in a web browser on behalf of the user, by directly manipulating its human-facing graphical user interface (UI).

Many everyday tasks we perform on the web are simple, yet repetitive and time-consuming. Filling out redundant administrative forms, visiting e-commerce websites in search of the best deal, or manually synchronizing different calendars are all tedious tasks that involve multi-step execution. Having autonomous web assistants that can follow instructions and execute such tasks on our behalf would empower users to focus on higher-level, higher-value decisions instead. They also represent a great opportunity to improve digital accessibility, by helping visually or otherwise impaired users who may find certain websites difficult to navigate. Lastly, assistants acting through a UI offer interpretability, since users can visually verify what the agent does on the screen, along with a wide compatibility, bypassing the need to develop specific application programming interfaces (APIs) to adapt existing software for the agentic era. We also note that web agent research is conveniently accessible to LLMs due to the textual modality of the UI, although most research on planning and reasoning with web agents is likely to translate to pixel-based UI agents.

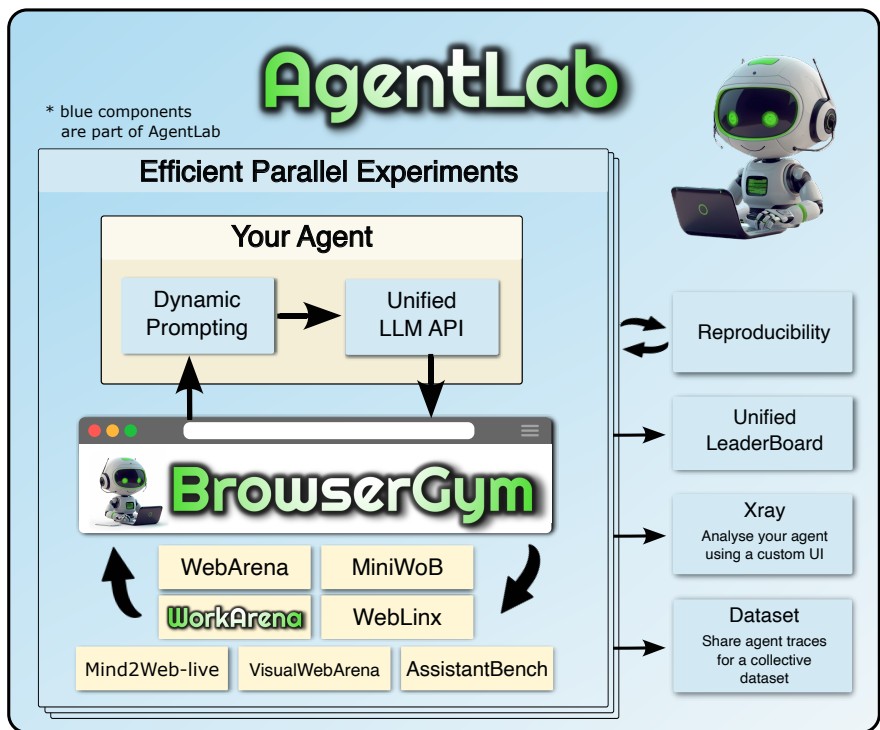

**Figure 1:** Overview of the ecosystem, including AgentLab, BrowserGym, and the web agent benchmarks.

The field of web agents has been particularly active in the last few years. Most recent works focus on evaluating the capabilities of state-of-the-art LLMs / VLMs at solving tasks of varying difficulty, which has led to an abundance of new web agent benchmarks (Zhou et al., 2024b; Koh et al., 2024a; Lù et al., 2024; Drouin et al., 2024; He et al., 2024; Boisvert et al., 2024; Yoran et al., 2024; Pan et al., 2024). Unfortunately, with this multitude of contributions also comes a multitude of diverging codes for evaluating web agents, which yields research silos with inconsistent practices and incompatible implementations. This fragmentation is detrimental to the fair comparison of existing web agent implementations across benchmarks, which compounds the cumbersome process of installing diverse setups, adapting data formats, and manually incorporating agents into each separate benchmark.

In this paper, we introduce the BrowserGym ecosystem, highlighted in Figure 1, which is meant to provide an open, easy-to-use, and extensible framework to accelerate the field of web agent research.[1] It consists of AgentLab, which offers the tooling for building web agents and running large-scale experiments, and a unified collection of

---

[1]It is important to note that this work is meant to provide a research framework, not a consumer product.

web agent benchmarks exposed through BrowserGym (Drouin et al., 2024), which provides the standard interface for web tasks and web agents. In its philosophy, the BrowserGym ecosystem intends to address the three following use cases: a) the design of new web agents, and their straightforward evaluation on a wide range of existing benchmarks; b) the design of new benchmarks, and the straightforward evaluation of existing web agents on them; c) the comparison of different models (LLMs / VLMs) in their ability to solve web agent tasks, by switching the backbone model in a state-of-the-art web agent implementation.

We summarize our contributions as follows:

- We expand the existing **BrowserGym**[2] library from Drouin et al. (2024) (Section 3), and we provide a **unification of existing benchmarks** (Section 4) proposed by the scientific community in BrowserGym, ranging from MiniWoB(++) (Shi et al., 2017; Liu et al., 2018) to more recent benchmarks such as AssistantBench (Yoran et al., 2024) and VisualWebArena (Koh et al., 2024a). Despite their differences, all benchmarks are made available through the same, unified BrowserGym interface.

- We introduce **AgentLab**[3] (Section 5), a set of tools to simplify parallel large-scale experimentation with agents over BrowserGym in a reproducible manner. It also comes with AgentXRay, a visual tool to introspect the behavior of agents on individual tasks. Finally, it provides reusable building blocks to accelerate the development of new agents.

- We conduct the first, large-scale **web agent experiment** (Section 6) showcasing the BrowserGym ecosystem. We compare the performance of six state-of-the-art LLMs / VLMs from the GPT-4 (OpenAI, 2024a), Claude 3.5 (Anthropic, 2024a) and Llama 3.1 families (Meta, 2024) across all web agent benchmarks available in BrowserGym. Among others, these results highlight an impressive performance for the new Claude-3.5-Sonnet model, which reaches an unprecedented success rate of 39.1% on the WorkArena L2 benchmark, to be compared with 8.5% for GPT-4o (2nd place). These results serve as first entries to the **BrowserGym leaderboard**.[4]

## 2 Background and Related Works

The field of autonomous agents has long been a major research interest both in academia and industry, with a recent surge in LLM-based agents (Wang et al., 2024a), motivated by the emergence of reasoning and planning capabilities in LLMs (Huang et al., 2024b; Wang et al., 2024c). While a large body of work focuses on software control via APIs (Yang et al., 2023a; Hao et al., 2024; Du et al., 2024), another line of research focuses on automating human-like interactions, by directly manipulating UIs on mobile devices (Li et al., 2020; Rawles et al., 2023; 2024), desktops (Xie et al., 2024; Bonatti et al., 2024), and websites (Furuta et al., 2023; Gur et al., 2023a; Kim et al., 2023; Drouin et al., 2024; Lù et al., 2024). This last category encompasses the field of web agents, which can automate software interaction even in environments without APIs, improve human productivity, and provide accessibility for users with disabilities. Next, we provide an overview of the recent literature related to web agents, which we divide into two parts: web agent benchmarks and web agent implementations.

### 2.1 Web Agent Benchmarks

Given the rapid pace of improvement in the field of Artificial Intelligence (AI), web agent benchmarks have been consistently evolving to evaluate more and more complex capabilities. Early benchmarks such as MiniWoB and MiniWoB++ (Shi et al., 2017; Liu et al., 2018) established foundational metrics for evaluating basic web interactions like form filling and button clicking in controlled environments. WebShop (Yao et al., 2022a) later introduced complex e-commerce scenarios, requiring agents to navigate product catalogs while considering user preferences and constraints. Mind2Web (Deng et al., 2023) further expanded this scope and released a dataset of over 2,000 human traces collected from open-ended tasks across 137 real-world websites, along with a structured approach to evaluate multi-step interactions to develop and evaluate generalist web agents. WebVoyager (He et al., 2024) proposed a live, online benchmark for evaluating web agents on real-world tasks from 15 popular

---

[2]`https://github.com/ServiceNow/BrowserGym`
[3]`https://github.com/ServiceNow/AgentLab`
[4]`https://huggingface.co/spaces/ServiceNow/browsergym-leaderboard`

websites. WebCanvas (Pan et al., 2024) later extended Mind2Web from a static dataset to an online benchmark called Mind2Web-Live, by proposing a key-node evaluation mechanism for web agents.

WebArena (Zhou et al., 2024b), along with its counterparts VisualWebArena (Koh et al., 2024a) and VideoWebArena (Jang et al., 2024), brought innovation by testing agents on live websites rather than simulated environments. This approach evaluated agents' abilities to handle real-world variability in website design and the dynamic nature of the content. VisualWebArena specifically addressed the role of visual understanding in modern web interfaces by introducing evaluation tasks that require interpreting and manipulating visual content across e-commerce websites, including product image comparison, visual search, and spatial reasoning about interface layouts. WebLINX (Lù et al., 2024) focuses on the instruction-following abilities of agents and provides a dataset of over 100k human interactions across 155 real-world websites. Building in the same direction, AssistantBench (Yoran et al., 2024) presents 214 realistic tasks requiring agents to navigate multiple websites to gather and synthesize information. MMInA (Zhang et al., 2024b) extends this further with 1,050 multimodal tasks that require agents to navigate between multiple websites, specifically evaluating their ability to handle both visual and textual information across naturally compositional tasks. In the same vein, GAIA (Mialon et al., 2023) proposes another question-answering benchmark that requires agents to navigate and combine information from diverse sources of the web to find the correct answer.

While these benchmarks focus on everyday websites, WorkArena and WorkArena++ (Drouin et al., 2024; Boisvert et al., 2024) introduced the first benchmark for web agents in the enterprise software setting, evaluating on common knowledge work tasks following realistic workflow. CRMArena (Huang et al., 2024a) proposes nine customer service tasks distributed across three personas: service agent, analyst, and manager, designed to evaluate the abilities of AI agents in CRM systems.

Broader evaluation frameworks like AgentBench (Liu et al., 2023b), AndroidWorld (Rawles et al., 2024), OS World (Xie et al., 2024), and Windows Agent Arena (Bonatti et al., 2024) incorporate web tasks as part of a more general agentic environment, providing insights into more general agent capabilities. These benchmarks explore the intersection of web, desktop, and mobile interfaces, recognizing the increasingly blurred boundaries between user interfaces. Addressing the need for safety evaluation, ST-WebAgentBench (Levy et al., 2024) introduces the first benchmark specifically focused on assessing web agents' compliance with organizational policies and safety requirements in enterprise settings.

## 2.2 Web Agent Implementations

Due to their immense potential, there has been a substantial effort in developing web agents. These agents differ in how they interact with the web environment, the backbone models, and the goal they aim to achieve, among others. Popular approaches either interact directly with HTML elements (Nakano et al., 2022; Deng et al., 2023; Gur et al., 2023b; 2024) or leverage vision-language models (Liu et al., 2023a; OpenAI, 2024a; Anthropic, 2024a; Meta, 2024), by grounding screenshots to web elements (He et al., 2024; Furuta et al., 2024; Zheng et al., 2024a; Koh et al., 2024a; Kil et al., 2024) or directly at the pixel-level (Shaw et al., 2023; Cheng et al., 2024; Gou et al., 2024). Beyond interaction methods, researchers have also explored ways to enhance agent reasoning capabilities through prompting or tuning. Previous works suggested specialized prompting (Sridhar et al., 2023; Zheng et al., 2024c; Sarch et al., 2024; Chi et al., 2024), planning and search architectures (Zhou et al., 2024a; Yoran et al., 2024; Zhang et al., 2024a; Gu et al., 2024; Koh et al., 2024b), and training methods (Gur et al., 2019; Nakano et al., 2022; Humphreys et al., 2022; Chen et al., 2023; Putta et al., 2024; Sodhi et al., 2024) to improve performance. Due to the recent success of LLM-based agents and the challenging reasoning required, agents often use prompting techniques such as chain-of-thought (Wei et al., 2023), tree-of-thought (Yao et al., 2023), ReAct (Yao et al., 2022b), and RCI (Kim et al., 2023). Wang et al. (2024a) provides a survey of LM-based agents.

Other works focus on introducing frameworks for web agent development, by simplifying access to evaluation datasets (Lhoest et al., 2021; UK AI Safety Institute, 2024; Ma et al., 2024), agent development (Xie et al., 2023), and interaction with live websites (Pan et al., 2024; Zheng et al., 2024b). It is thus important to focus on standardizing web agents research, aiming to simplify future work, and introducing research best practices—standard evaluation with statistical power across benchmarks, simple ecosystem-growth enabling open-source research, and reproducible experiments at scale. Such a framework is needed not only for the development of the next generation of open-source web agents but also for general tasks that require web access,

including the development of autonomous coding agents (Wang et al., 2024b). To address those important points, we developed BrowserGym and AgentLab, which we describe in detail in the subsequent sections.

## 3    BrowserGym

BrowserGym provides a unified interface for web agents, i.e., conversational assistants that can use a web browser as a tool. As showcased in Figure 2, both the user and the agent have access to a chat where they can exchange messages, and a browser in which they can perform actions, such as typing text, clicking buttons, or opening tabs. The user typically writes instructions in the chat, and the agent tries to follow them by navigating across pages, performing UI actions, extracting information, and writing back messages in the chat.

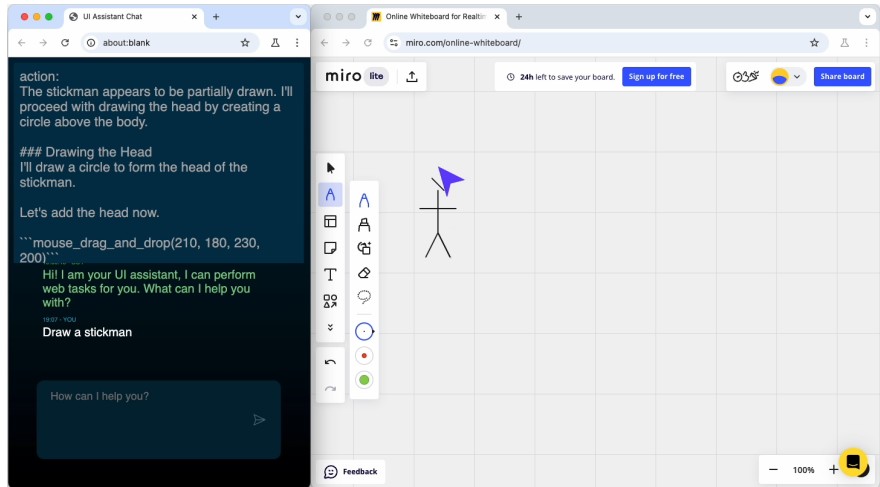

**Figure 2:** A rendered BrowserGym environment, with the chat on the left and the web browser on the right.

**Implementation**    From the agent's perspective, this interaction loop can be conceptualized as a Partially Observable Markov Decision Process (POMDP), where the environment (chat and browser) provides the current observation and reward, and the agent produces the next action. BrowserGym environments follow the standard OpenAI Gym API (Brockman et al., 2016)(Towers et al., 2024) and are made available in Python through the gymnasium[5] interface (Figure 3). Internally, BrowserGym relies on the Chromium[6] browser and the Playwright[7] library to automate browser interactions.

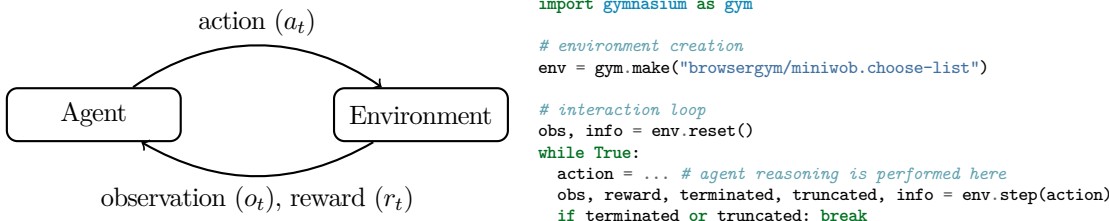

**Figure 3:** The BrowserGym interaction loop: (left) theoretical model, (right) concrete Python API.

### 3.1    Extensive Observation Space

BrowserGym provides a rich observation space, with its main components being the task's goal (or chat history), the list of open tabs, and the current page's content.

---

[5] https://gymnasium.farama.org/
[6] https://www.chromium.org/Home/
[7] https://playwright.dev/python/

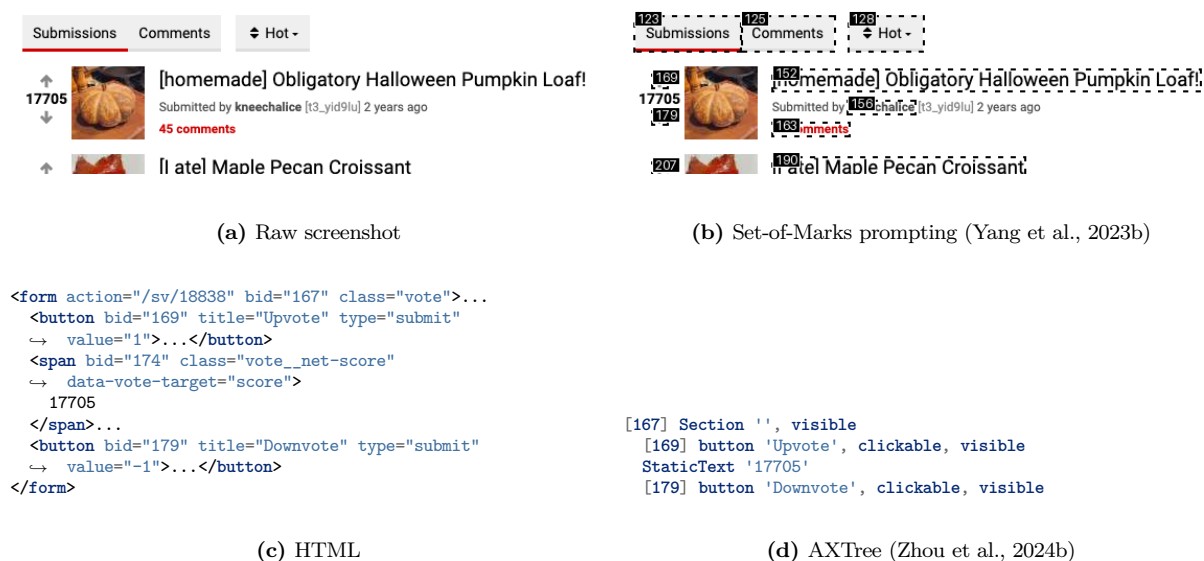

**(a)** Raw screenshot

**(b)** Set-of-Marks prompting (Yang et al., 2023b)

```
<form action="/sv/18838" bid="167" class="vote">...
  <button bid="169" title="Upvote" type="submit"
  ↪  value="1">...</button>
  
    17705
  ...
  <button bid="179" title="Downvote" type="submit"
  ↪  value="-1">...</button>
</form>
```

```
[167] Section '', visible
  [169] button 'Upvote', clickable, visible
  StaticText '17705'
  [179] button 'Downvote', clickable, visible
```

**(c)** HTML

**(d)** AXTree (Zhou et al., 2024b)

**Figure 4:** Different visual and text representations of the current page from BrowserGym.

**Structured page description** BrowserGym provides the raw DOM (Document Object Model; Web Hypertext Application Technology Working Group, 2024) and the AXTree (accessibility tree; World Wide Web Consortium, 2017) as structured representations of the current page, which are extracted using the Chrome Developer Protocol (CDP). They are both provided as-is (`dom_object`, `axtree_object`) with only minimal alterations such as the injection of unique identifier attribute (`bid`, see next paragraph). It is then left for the web agent implementation to decide how to further process and format these observations, for example by filtering irrelevant elements or attributes and converting the DOM or AXTree to a suitable text format for LLM-based text agents (see Figures 4c and 4d). Utility functions that convert these objects to basic text representation are also provided, for implementing simple agents quickly.

**Extra element properties** BrowserGym enriches each HTML element in the current webpage with extra information to facilitate web agent interactions. One key property is the BrowserGym ID (`bid`), a uniquely-generated, element-wise identifier injected into the page's DOM and AXTree, allowing for unambiguous interaction with the elements of the page. To add information about the visual rendering of the page, BrowserGym also provides for each element its bounding box coordinates (`bbox`) in the form of a (left, top, width, height) 4-tuple, as well as its visibility ratio[8] (`visibility`) in the form of a scalar between 0 and 1. Finally, BrowserGym also provides a Set-of-Marks boolean indicator (`set_of_marks`) reproduced from He et al. (2024), which indicates if an element's bounding box should be overlayed on the screenshot to enable Set-of-Marks prompting (Yang et al., 2023b; He et al., 2024). (see Figures 4a and 4b).

**Screenshot** While the DOM, AXTree, and extra element properties provide a rich description of the page's content, they do not give the full picture of how the page is being rendered in the browser, i.e., overlapping elements, background colors, image contents, fonts, etc. Therefore, BrowserGym also provides the raw `screenshot` of the current page as part of the observation, in the form of an RGB image. Again, it is up to the agent implementation to decide how to process and use this visual information. Note that the `bbox` coordinates in the extra element properties map exactly to pixel coordinates in the screenshot, which makes it extremely easy for agents to overlay the bounding boxes and identifiers of relevant elements on the screenshot as in Set-of-Mark prompting (Yang et al., 2023b; He et al., 2024).

**Open tabs** In BrowserGym, tasks can use multiple tabs if needed. While the main focus is on the current page (screenshot and page description), agents are also made aware of all the open browser pages, which include tabs and pop-up windows. BrowserGym's observations therefore include a list of all the pages' URLs and

---

[8]https://developer.mozilla.org/en-US/docs/Web/API/Intersection_Observer_API

titles (`open_pages_urls`, `open_pages_titles`), as well as an index to distinguish the currently active page (`active_page_index`).

**Goal and chat messages**  In BrowserGym web agents can extract the user instructions in two ways, either explicitly from the task goal (`goal_object`) or implicitly from the chat (`chat_messages`). Both consist of a list of text and/or image messages. This duality is offered to facilitate compatibility with legacy web agent and web task implementations, which do not account for an interactive chat in their design. Note however that BrowserGym will always inject the goal of a non-interactive task in the chat to mimic user interaction, but will not inject further chat messages received during a task execution back into the goal. While, to this date, all web agent benchmarks in the literature are non-interactive, it is preferable to implement web agents using `chat_messages`, at the very least to be able to record live, interactive demos.

**Error feedback**  Finally, BrowserGym observations always include the error message of the last executed action, if any (`last_action_error`). Typical errors include trying to interact with an element that is hidden or deactivated, which results in a Playwright exception with an error message. Such exceptions do not break the interaction loop but are instead captured, and are sent back to the agent as part of the next observation. This immediate negative feedback gives web agents a chance to quickly self-correct in the next step (See Figure 5).

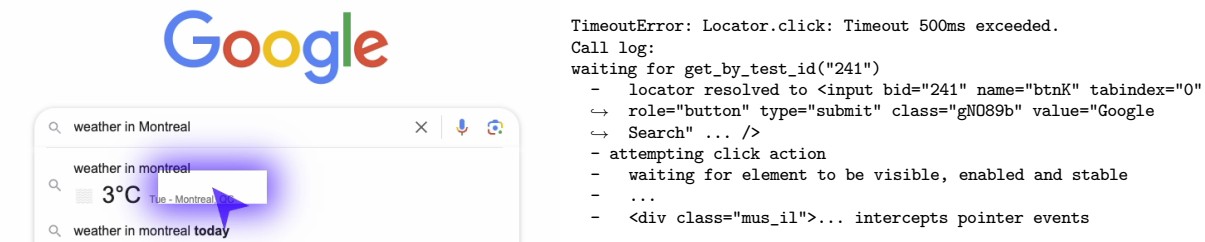

**Figure 5:** Example of an agent trying to click on the "Google Search" button, hidden behind the auto-complete menu. This action results in a Playwright error that is returned in the next observation's `last_action_error` field.

```python
# locate and click the upvote button with Playwright
button = page.locator("form").filter(has_text="17705")
↪    .get_by_role("button").first
button.click()

# or, do it in Javascript
page.evaluate("document.querySelectorAll('form')\
  .find(form => form.textContent.includes("17705"))\
  .querySelector('button').click();")

# send a chat message
send_message_to_user("The \"Upvote\" button has been clicked")
```

**(a)** Raw Python code

```python
# click using bid
click("169")

# or, focus and press enter
press("169", "Enter")

# or, click using coordinates
mouse_click(x=121, y=197)

# send a chat message
send_msg_to_user("The \"Upvote\" button has been clicked")
```

**(b)** High-level primitives

**Figure 6:** Different BrowserGym actions for clicking the "Upvote" button of the first post in Figure 4. While raw Python code (a) enables expressive actions using Playwright or Javascript, it often relies on some knowledge of the page's HTML structure. Using BrowserGym's high-level action primitives (b) is more restrictive but offers more control over what the agent can do and how it should interact with the browser.

## 3.2   Expandable Action Space

The fundamental action space of BrowserGym is raw executable Python code (Figure 6a). Agents have access to a Playwright `page` object along with two functions `send_message_to_user(text)` and `report_infeasible(reason)`. These respectively allow agents to interact with the browser, and the chat, and terminate the task. This design choice offers a powerful, unrestricted action space for researchers to play with, but also exposes web agent users to vulnerabilities and safety risks due to the possibility of executing arbitrary, untrusted code. To mitigate this, BrowserGym offers a control mechanism to further restrict the action space available to web agents, through the use of an `action_mapping` function.

**Action mapping**  In BrowserGym, users can instantiate environments with an optional `action_mapping` function, which the environment uses to convert input actions into executable Python code. It is then easy to define and experiment with new, custom action spaces in BrowserGym, for example by allowing a pre-defined list of actions in JSON or in command-line format, which are parsed and converted to safe, trusted Python code. In case of a conversion error (e.g., parsing errors), any Exception message raised by `action_mapping()` is caught and fed back to the agent as an error feedback.

**High-level action set**  By default, BrowserGym environments use a pre-defined action mapping (Figure 6b), which converts a set of high-level action primitives in the form of function calls into actual Playwright code to be executed. An exhaustive list of these primitives can be found in Appendix A, Table 3. The resulting action set is encapsulated in the `HighLevelActionSet` class, which provides a `to_python_code()` method for the environment (the action mapping), and a `describe()` method for the agent, which builds a textual description of the available high-level primitives and how to use them (Appendix A, Figure 10).

### 3.3  Extensibility: create your own task

```python
class MyNewTask(AbstractBrowserTask):

    def setup(self, page):
        page.goto("http://google.com")
        return "Which year was einstein born?"

    def validate(self, page, chat_messages):
        # stop after first answer
        done = chat_messages[-1]["role"] == "assistant"
        reward = "1879" in chat_messages[-1]["message"]
        message = "That's correct" if reward else ""
        return reward, done, message

register_task("mynewtask", MyNewTask)

# available as a gym environment
env = gym.make("browsergym/mynewtask")
```

**Figure 7:** Pseudo-code for creating a simple web task in BrowserGym, and the corresponding rendering.

BrowserGym minimizes the effort needed for practitioners to implement new web tasks. Doing so only requires writing the logic of the task, which takes the form of a `setup()` and a `validate()` method.

- **Setup** Starting from a blank `page` object, this method must bring the browser to the starting point of the task, and return the goal the web agent must achieve. This will typically involve logging into a system, creating data entries if needed, and navigating to a task-specific starting URL. Then, the method must return a goal which can either be a raw string as in Figure 7 or a list of OpenAI-style messages[9] that can include text and images, to allow for vision-conditioned goals (e.g., "Find a pair of shoes that look like this image").

- **Validate** Called after each agent action is executed, this method must check whether the agent has completed the task. It has access to the current state of the browser through the active `page`, as well as the history of `chat_messages` which contains all previous user and assistant messages. Typical validation steps involve looking at the content of the current page, searching for database entries and updates, or reading assistant messages in the chat in search of a correct answer. At the end, this method must produce a scalar `reward` for the agent, a boolean `done` flag that indicates task termination, and an optional `message` to be added to the chat, to simulate user interaction.

## 4  Unification of Web Agent Benchmarks

We bring 3 new web agent benchmarks to BrowserGym, namely WebLINX (Lù et al., 2024), VisualWebArena (Koh et al., 2024a) and AssistantBench (Yoran et al., 2024). With these, BrowserGym currently supports six popular

---

[9]https://platform.openai.com/docs/api-reference/messages

**Table 1:** Overview of the Web Agent benchmarks currently available in BrowserGym. To quantify the task diversity, we report the number of task templates and their seed diversity, e.g. in WorkArena, some task templates can have thousands of different configurations. On the other hand tasks in, e.g., WebArena are deterministic.

| Benchmark | # Task Templates | Seed Diversity | Max Steps | Multi-Tab | Web Backend | Reference |
|---|---|---|---|---|---|---|
| MiniWoB(++) | 125 | medium | 10 | no | self-hosted pages | Shi et al. (2017) Liu et al. (2018) |
| WebArena | 812 | none | 30 | yes | self-hosted docker | Zhou et al. (2024b) |
| VisualWebArena | 910 | none | 30 | yes | self-hosted docker | Koh et al. (2024a) |
| WorkArena L1 | 33 | high | 30 | no | ServiceNow demo instance | Drouin et al. (2024) |
| WorkArena L2 | 341 | high | 50 | yes | ServiceNow demo instance | Boisvert et al. (2024) |
| WorkArena L3 | 341 | high | 50 | yes | ServiceNow demo instance | Boisvert et al. (2024) |
| WebLINX | 31586 | none | 1 | no | none (static dataset) | Lù et al. (2024) |
| Assistant Bench | 214 | none | 30 | yes | world wide web | Yoran et al. (2024) |

web agent benchmarks, listed in Table 1. Each benchmark consists of a set of BrowserGym tasks, accessible as a BrowserGym environment through the gymnasium interface (Figure 8). Any task from any of these benchmarks can then be executed using the same code base, through the same observation/action API, and any web agent implemented for one benchmark can readily be evaluated on another.[10] The advantages are the following:

- **Breaking silos**. Unifying benchmarks encourages the development of general-purpose web agents. While benchmark-specific solutions have their own merit (Kim et al., 2023; Sodhi et al., 2024) and correspond to real use cases (e.g., a company developing a web agent for an internal website), we believe that the pursuit of generic solutions is more likely to bring long-term, impactful innovation.

- **Accelerating research**. Proposing a unified interface for web agents will also encourage the development of new, compatible benchmarks, stimulate progress in the field, and accelerate research. For example, a new BrowserGym-compatible security benchmark could easily evaluate all existing web agent implementations, without the need to spend time and effort adapting agent codes.

- **Reducing noise**. By evaluating across benchmarks, practitioners can collect stronger statistical signals for testing design choices and scientific hypotheses. Does using screenshots improve the performance of web agents? Is LLM-A a better web agent than LLM-B? Evaluating on a single benchmark with its own biases and specificities only provides a partial answer, while cross-benchmark evaluation gives a wider picture.

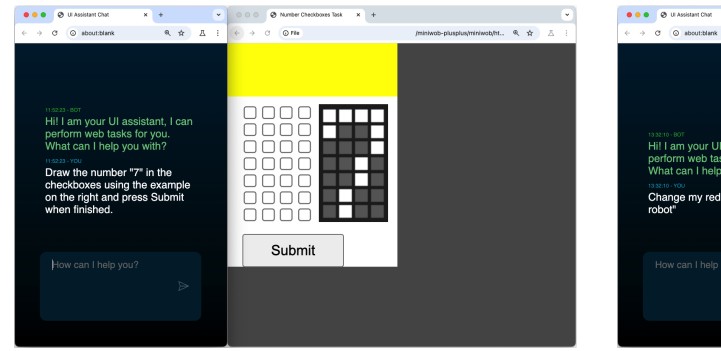
gym.make("browsergym/miniwob.number-checkboxes")

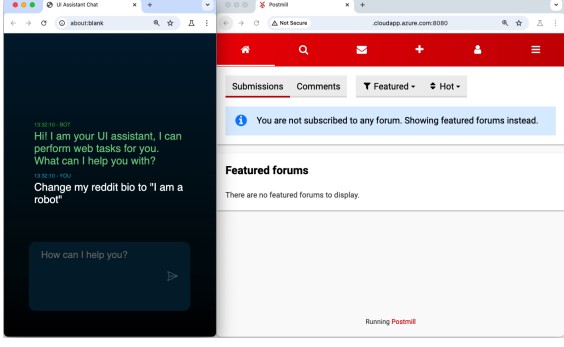
gym.make("browsergym/webarena.399")

**Figure 8:** Two tasks from the MiniWoB and WebArena benchmarks, accessible through the same API.

---

[10]BrowserGym also suggests a unified API for agent creation which streamlines implementation efforts (Section 5.5).

**Task coverage** Benchmarks currently available in BrowserGym cover a broad range of tasks that involve manipulating synthetic UIs (MiniWoB), performing everyday tasks on replicas of real-world shopping, social media, and software development platforms (WebArena, VisualWebArena), executing realistic workflows on an enterprise platform (WorkArena), running time-consuming searches on the open web (AssistantBench), and replicating human interaction traces (WebLINX). For a detailed presentation of each of these benchmarks, we refer the reader to Appendix B. Additional benchmarks currently considered for integration include GAIA (Mialon et al., 2023), WebVoyager (He et al., 2024), WebShop (Yao et al., 2022a) and Mind2Web-live (Pan et al., 2024).

**Metadata** Each BrowserGym benchmark comes with its own metadata, which lists all the available tasks along with their benchmark-specific attributes. For example, the metadata of WorkArena (Drouin et al., 2024; Boisvert et al., 2024) contains the level and the category of each task, which is useful for analyzing the performance of web agents on specific subsets of tasks. The metadata also proposes a default train/test split for each benchmark, and an optional dependency graph between tasks, which indicates a (partial) order in which tasks should be executed to avoid inconsistencies when evaluating agents (e.g., for the WebArena (Zhou et al., 2024b) and VisualWebArena (Koh et al., 2024a) benchmarks).

**Suggested evaluation parameters** On top of the metadata, each BrowserGym benchmark has its own suggested action set, suggested number of seeds per task, and suggested maximum number of steps to be used for evaluation. These suggested parameters are not mandatory but provide a preferred way to evaluate web agents on each benchmark, following the recommendations of the benchmark authors. For example, the MiniWoB(++) benchmark (Shi et al., 2017; Liu et al., 2018) is not intended to be used in a multiple tab setting, hence does not need the `new_tab()` or `close_tab()` action primitives. The AssistantBench benchmark (Yoran et al., 2024) on the other hand does not implement task randomization, hence does not need to be evaluated with more than one seed per task.

**Backend preparation** Some benchmarks require resetting the backend web server to a specific state before the next agent can be evaluated on it. It is the case for WebArena and VisualWebArena, which require restoring docker containers to their initial state so that any alteration performed by a previous agent (e.g., changing the user's bio description) is wiped off for the next agent to start in a clean, fresh server state. BrowserGym automates this step with a convenience `prepare_backend()` routine, which checks the backend configuration (server URL, credentials, etc.) and runs any required reset or setup automatically.

# 5 AgentLab

As depicted in Figure 1, AgentLab is a set of tools to simplify experimenting with agents over BrowserGym. In the high-level API, we define the `Study` object to organize the experiments across multiple agents on a benchmark. The study is materialized on disk and can be launched to run in parallel across multiple processes (Section 5.2). Once a study is completed it can be examined using AgentXRay (Section 5.3), a custom UI made to introspect the behavior of agents on individual steps of each task in the study. Due to the challenges of reproducing experiments in dynamic environments, we also implement capabilities to help reproducibility (Section 5.4). Finally, we also provide building blocks to simplify creating new agents (Section 5.5).

## 5.1 Launching experiments

Once an agent is implemented (See section 5.5), it can be evaluated by creating a `Study` object and running it:

```
study = make_study(
    benchmark="miniwob",  # or "webarena", "workarena_l1" ...
    agent_args=[AGENT_4o_MINI],
)
study.run(n_jobs=5)
```

`Study` objects are in charge of setting up the experiment and running and saving the reproducibility checks. They also manage the relaunching of failed experiments. In such a broad range of environments, it is frequent to experience system failures such as a server not responding, edge-case bugs in third party libraries, or rate

limits from LLM servers. Thus, it is crucial to be able to relaunch failed experiments. The method `Study.run` is designed to relaunch up to 3 times the failed tasks, but a manual relaunch can also be done as follows:

```python
study = Study.load("/path/to/your/study/dir")
study.find_incomplete(include_errors=True)
study.run()
```

### 5.2   Parallel experiments

Evaluating a single agent on a benchmark with a reasonable number of seeds per task yields several hundred episodes. When exploring multiple agent configurations this can lead to several thousand episodes in a single study. Hence, it is crucial to parallelize efficiently the execution of these episodes.

We use a multiprocess approach with the possibility to use `ray` (Moritz et al., 2018) or `joblib`[11] as a backend. Since most of the compute requirements are outsourced to servers (LLM API, TGI server, or web server), the actual compute requirement of the job is relatively small and we can launch up to 20 tasks in parallel on a single laptop or even 50–100 tasks on a server machine with many CPU cores and large memory. With such parallelization, the rate limits of API-based LLMs are usually the bottleneck. If more parallelism is required, `ray` can connect multiple machines into a single pool of workers. We refer the reader to Appendix G for further details regarding hardware and runtime.

**Task dependencies**   Benchmarks like (Visual)WebArena have task dependencies to ensure that the execution of a task will not corrupt the instance and prevent it from properly solving other tasks. The `ray` backend is capable of handling such dependencies,[12] but unfortunately this also greatly limits the potential parallelism to 2–4 tasks in parallel for the (Visual)WebArena benchmarks. Also, if the study is created through `make_study`, AgentLab ensures the sequential execution of all agents in the study with a proper instance reset between each evaluation.

### 5.3   AgentXRay

To facilitate deeper analysis of agent behavior, AgentLab includes AgentXRay (not be confused with the `ray` parallel backend), a Gradio-based[13] interface for diving deep into the logged traces. A visual example of this interface is shown in Figure 9. After selecting the trace of interest AgentXRay displays the profiling, the goal, the different parts of the observation, the action taken, and the prompts, providing a comprehensive view of the agent's decisions and interactions. This interface allows users to visualize the step-by-step decision-making process of an agent, helping them gain valuable insights and refine their agent for better performance. This is particularly useful for identifying areas where the agent may be struggling, as it allows researchers to pinpoint the exact moment when an incorrect or suboptimal decision is made.

### 5.4   Reproducibility

Several factors can influence the reproducibility of results in the context of evaluating agents on dynamic benchmarks. Having a framework robust to the noisy nature of web tasks is necessary. We first discuss core factors that can lead to discrepancies in web agent evaluations and then present the corresponding crucial features that AgentLab implements to mitigate these discrepancies.

#### 5.4.1   Factors affecting reproducibility

**Software version**   Different versions of Playwright or any package in the software stack could influence the behavior of the benchmark or the agent. Fixing package versions can help, but it would prevent bug fixes, security updates, and new features.

**API-based LLMs silently changing**   Even for a fixed version, a commercial LLM may be updated, for example, to incorporate the latest web knowledge.

---

[11] https://github.com/joblib/joblib
[12] `joblib` cannot handle dependency graphs.
[13] https://www.gradio.app

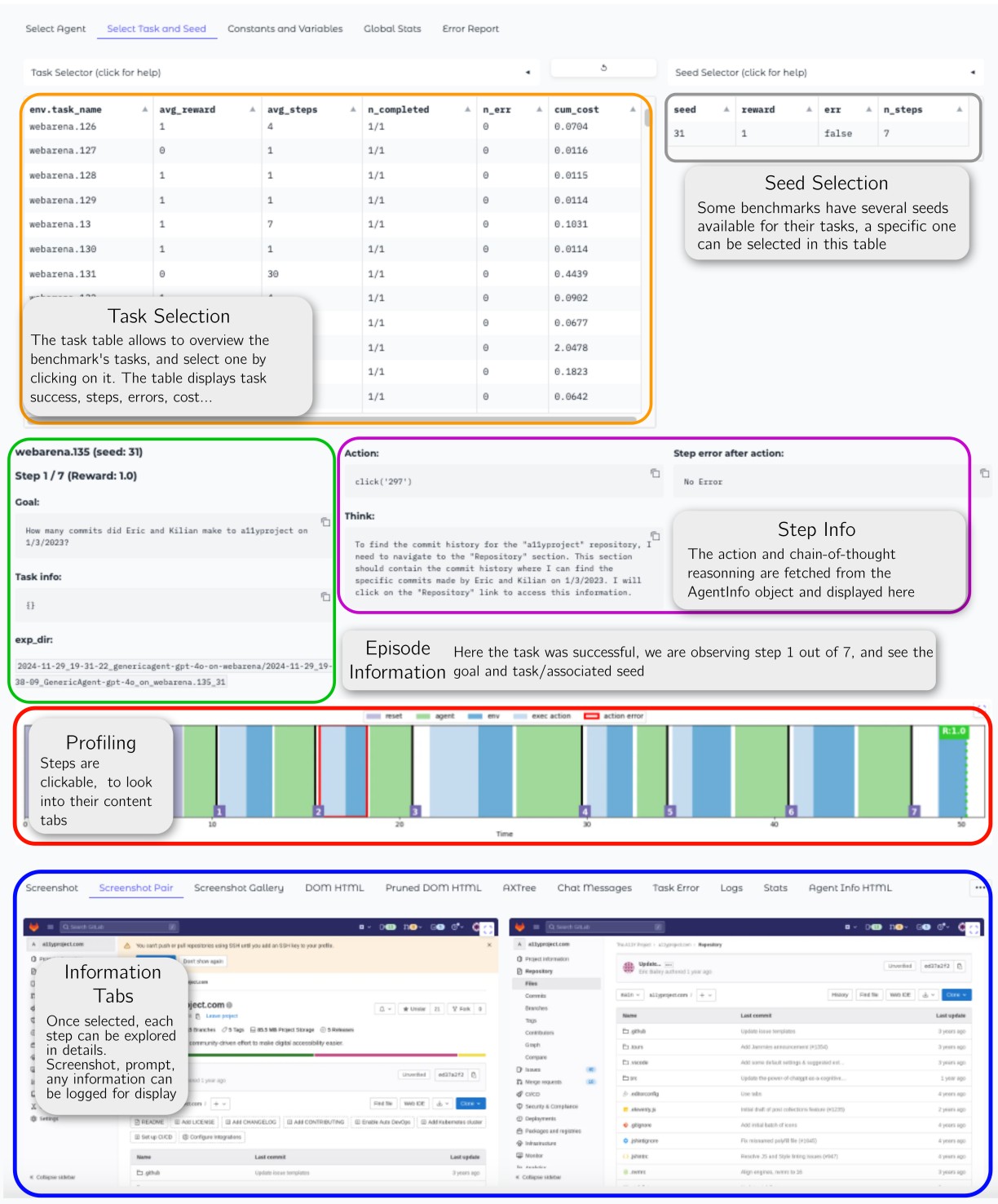

**Figure 9:** Visual rendering of AgentLab's XRay interface. AgentXRay serves as a powerful analysis tool to inspect and work on BrowserGym agents.

**Live websites**  Agents that interact with live web content are subject to frequent changes in website design, content availability, and default language settings, which can vary across countries or regions. Websites may update layouts, navigation, or features without notice, disrupting an agent's performance. Additionally, reliance on dynamic or frequently updated sites introduces unpredictability, as content and structure can shift over time, making consistent evaluation more challenging.

**Stochastic Agents**  LLMs are non-deterministic, but this is less concerning since it is independent and identically distributed (IID) noise, and setting the temperature of the LLM to 0 can reduce most stochasticity.

**Non-deterministic tasks**  Similarly, many tasks contain some level of stochasticity. However, changes should be minimal for a fixed seed.

### 5.4.2 Reproducibility Features

**Standard observation and action space**  One core feature of BrowserGym is to provide a standardized observation and action space. While it allows for customization, each benchmark is defined with its own default action space. Similarly, the observation space of agents in AgentLab comes with a default way to represent the DOM, the AXTree, and other components.

**Package versions**  The `Study` class contains a dictionary of information about reproducibility, including benchmark version, package version, commit hash, os version, and time stamp. By keeping track of these variables, one can isolate package versions, interactions, or system effects that can impact the performance.

**Reproducibility Journal**  A call to `study.append_to_journal()` automatically uploads your results to `reproducibility_journal.csv`. The goal of this journal is to keep a detailed track of previous results. As mentioned in Section 5.4.1, because of API changes or instance changes, results can vary overtime. This makes it easier to populate a large number of reference points. The journal aims to track changes and log enough information to allow anyone to reproduce as closely as possible an experiment. It stores information such as agent name, BrowserGym and AgentLab version, benchmark name and metadata, experiment performances, etc.

**Reproduced results in the leaderboard**  For experiments that are reproducible, we encourage users to try to reproduce the results and upload them to the leaderboard. There is a special column containing information about all reproduced results of an agent on a benchmark. It is displayed as a range of scores alongside the original score on the leaderboard. The leaderboard, together with the Reproducibility Journal, helps monitor changes in benchmarks over time and serves as a source of motivation for the community.

**ReproducibilityAgent**  The base agent `GenericAgent` (detailed in section 6) comes along with an agent called `ReproducibilityAgent`. Running the latter on an existing study generated from `GenericAgent` will re-execute the same action sequence on the same task seeds. A visual diff of the two prompts is then displayed in the AgentInfo HTML tab of AgentXray. The user can inspect what changes occurred between the two executions. While the two executions may diverge rapidly if there are some differences at the start of the episode, the diffs between the first few steps are typically insightful on what has changed since the last execution. This allows one to monitor the modifications in live benchmarks that may be subject to changes over time.

## 5.5  Extensibility: create your own agent

At the core, BrowserGym and AgentLab are designed to have a minimalist API for implementing new agents. Only a few key components need to be implemented in the `Agent` class:

```python
class MyAgent(bgym.Agent):
    def __init__(self):
        # define an action_set to decode actions
        self.action_set = bgym.HighLevelActionSet()

    def get_action(self, obs):
        # the core of the agent reasoning goes here
        return action, agent_info
```

The main part is the `get_action` method, where one implements the agent's logic. It returns an action as a string to be interpreted by BrowserGym, based on the chosen `action_set`. The `get_action` method can also return `agent_info`, an object of type `AgentInfo` containing various informations for later inspection of the agent's behavior.

**AgentArgs**  On top of coding an agent, the user is also responsible for coding `AgentArgs`. This is essentially a data structure which the `make_agent` function uses for instantiating an agent. Doing so makes it easy and reliable to pickle experiment information to disk or send it across processes. AgentLab also provides a few other methods for better integration with the framework. For example:

```python
@dataclass
class MyAgentArgs(agentlab.AgentArgs):
    use_chain_of_thought: bool = True
    use_html: bool = False

    def make_agent(self): -> bgym.Agent
        return MyAgent(self.use_chain_of_thought, self.use_html)

    def set_benchmark(self, benchmark):
        # Optional method to set benchmark-specific flags
        if "miniwob" in benchmark.name:
            self.use_html = True
```

Once this is implemented, the user can evaluate their agent as presented in Section 5.1. Note: the class definition of e.g., `MyAgentArgs` needs to be saved in a module that can be imported from the `$PYTHONPATH` for proper unpickling.

**Dynamic Prompting**  While most modern LLMs can digest more than 100,000 tokens, AXTree can still be larger than 100,000 tokens, with the raw HTML/DOM counting more than 1,000,000 tokens. Additionally, research projects fine-tuning open-source LLMs are often limited to very small context lengths, e.g., 8,000. To manage this, our prompting tools are designed to fit the prompt to the desired length without having to naively truncate the prompt from the end, which would almost always cut crucial information such as examples. This is done using the `DynamicPrompting` class, where the user can easily configure which part of the BrowserGym observation space should be included in the prompt. Next, the length of the prompt is dynamically shrunk to fit a desired size. The `fit_token` method iteratively calls the shrink method recursively on each component of the prompt. Default shrinking strategies are implemented, such as reducing the number of previous steps included in the history, or truncating the page from the bottom. Users can also implement their own shrinking strategies. See Figure 11 for a simple example agent using dynamic prompting.

### 5.6  Extensibility: Plug your own LLM / VLM

AgentLab also offers a base class for LLMs. This allows users to plug their models into any existing agent by just sub-classing our objects. More precisely, we provide a `BaseModelArgs` class, to configure and log LLM information, as well as instantiating the model, and a `AbstractChatModel` class, which just needs a `__call__` method defined to query the LLM. AgentLab comes with different model objects readily available, such as a class for OpenAI models, Azure OpenAI models, or OpenRouter[14] models. Using our chat model classes allows us to add some useful tools such as a cost and token tracker that can automatically be added to the `AgentInfo`.

## 6  Experiments

This section showcases large-scale experimentation that leverages the BrowserGym ecosystem and evaluates the capabilities of several modern LLMs and VLMs in performing web tasks. We focus on two objectives:

- First, showcasing how AgentLab, integrated with BrowserGym, facilitates the creation and management of experiments with web agents.

- Second, assessing the current performance levels of various state-of-the-art models in tackling these benchmarks.

---

[14]https://openrouter.ai/

The results of this experiment provide an interesting overview of the strengths and limitations of the evaluated models, offering insights into axes where they require improvements. The results are consolidated into an online leaderboard, allowing for comparisons and updates as new models and benchmarks emerge.

## 6.1 Setup

We evaluate AgentLab's default agent, `GenericAgent`, on all available benchmarks in the BrowserGym ecosystem. `GenericAgent` employs a modular prompting mechanism that allows it to utilize various tools, such as Chain-of-Thought (CoT) reasoning, screenshots with Set-of-Marks (SoM), a memory mechanism, self-criticism, and in-context examples to enhance its responses. The configuration determines both the agent's behavior and the observation format, including options like using HTML or AXTree representations, retaining a history of observations and past actions, incorporating screenshots or Set-of-Marks images, or displaying additional HTML information from the page. The `ExampleAgent` shown in Figure 11 in the Appendix gives a simplified overview of our `GenericAgent`.

Along with this dynamic prompting feature, `GenericAgent` implements a retry functionality to overcome LLM side issues or parsing errors. In the case of a parsing error, the LLM is re-prompted and gets 4 attempts to produce a parsable answer. After 4 consecutive parsing errors, the task is considered a failure.

Our experiments use the same agent configuration as the WorkArena++ benchmark (Boisvert et al., 2024), with the addition of the `use_think_history` setting which gives the agent access to its entire chain-of-thought history throughout its execution, similarly to Putta et al. (2024). Intuitively, this allows the agent to remember previous interactions and outcomes, and make more informed and consistent decisions. An example of a resulting prompt is showcased in Appendix C.

We evaluate AgentLab's `GenericAgent` using a representative subset of state-of-the-art large language models, which includes:

- **GPT-4o** and **GPT-4o Mini** (OpenAI, 2024a) using the Azure OpenAI API, as some of the best models available on the market. We use respectively checkpoints 2024-08-06 and 2024-07-18.

- **o1 Mini** (OpenAI, 2024b) using the OpenRouter API, to test out the performances of test-time reasoning-focused models. We use the 2024-09-12 checkpoint for this model.

- **Claude 3.5 Sonnet** (Anthropic, 2024a) using the OpenRouter API, as the best-performing model on our benchmarks. We use the 2024-10-22 checkpoint, most recent to this day.

- **Llama-3.1** 70B and 405B (Meta, 2024) using the OpenRouter API, as representatives of the open-weights LLM community, to assess their performance relative to their proprietary counterparts and identify improvement axes.

As our evaluation metric, we use the task success rate, i.e. the percentage of tasks that were successfully completed by the agent, and compute the standard error ($\sigma/\sqrt{N}$). We evaluate our agent with all the previously mentioned LLMs, on all benchmarks, with some exceptions. On WorkArena L3, except for Claude, we report results obtained in Boisvert et al. (2024) as they are similar agents and we have strong expectations that the score would not change, leading to significant resource savings. On VisualWebArena, we only run the LLMs that support multi-modal inputs. Out of fairness between models, we do not use visual inputs for the other benchmarks, as they do not require it.

Table 5 in the Appendix shows the settings used for each benchmark and the number of evaluation tasks for each. Indeed the number of tasks (or episodes) depends on the number of seeds (MiniWob, WorkArena L1), or the split used. We use the test splits for WebLINX and AssistantBench. Finally, WorkArena L2 and L3 offer their own curricula, amounting to 235 tasks each.

## 6.2 Results

We present the results obtained using various open-source and closed-source LLMs as `GenericAgent`s across standard web agent benchmarks in Table 2. We observe that Claude obtains the best performance across the majority of the benchmarks, with o1-Mini falling second. Claude surprisingly improves substantially over the

previously unsolved WorkArena-L2 benchmark and obtains an average task success rate of 39.1%. One explanation for this impressive result could be Claude's specific training for computer use (Anthropic, 2024b), which bears a close resemblance to using a web browser. We also observe that Llama-3.1 70B obtains very close performance to GPT-4o Mini, and Llama-3.1-405B surpasses GPT-4o-Mini significantly on numerous benchmarks. While beaten on most complex reasoning benchmarks, 405B still obtains more than decent performances overall, which comes as a promising omen for the open-source community.

We also observe an improvement in performance for GPT-4o compared to the performance measured by Boisvert et al. (2024), which goes from 23.5% to 31.4% on WebArena, and from 3.8% to 8.5% on WorkArena L2. Since they were using a very similar web agent implementation but an older model checkpoint, this suggests that the reasoning capabilities of GPT-4o have greatly increased thanks to the additional training performed for this new checkpoint. Another option to explain this improvement might be the public availability of those benchmarks and their support, meaning parts of their content might now be integrated into some form in the LLM pre-training / fine-tuning datasets.

One interesting point is that our agent seems to underperform on AssistantBench, a web-based question-answering benchmark. Our best agent attains a score of 6.9%, which is a very low score compared to the current high score of 27% on the AssistantBench leaderboard. Our agent is not specifically optimized for information retrieval tasks, which are the primary focus of the AssistantBench benchmark. Instead, our API and agent are designed to be broadly applicable across a wide range of web-based tasks, prioritizing generality over specialization.

## 6.3 Errors

When evaluating AI agents, errors can generally be classified into a few broad categories. **Navigation Errors** occur when the agent struggles to locate the correct page or information, often due to poor search queries or navigation mistakes. **Form Handling Errors** involve incorrect data entry, such as formatting issues or failure to recognize submission failures. **Task Understanding Errors** arise when an agent misinterprets unclear instructions, leading to irrelevant or incorrect actions. **Stuck Behavior** happens when an agent repeats the same failing action without adapting to the situation. **Information Extraction Failures** occur when the agent successfully navigates to the right location but fails to correctly extract, interpret, or return the necessary data. Finally, **External Errors** stem from technical issues such as API failures, network errors, or system crashes that hinder the agent's performance. While these categories are not definitive, they offer a useful framework for diagnosing weaknesses and improving AI agent reliability.

Given the scale of our experiments, we only provide a qualitative overview of the errors encountered by the agent. However, we release the full traces of our experiments and let to the reader the possibility to skim through the data using AgentXRay. Appendix H showcases a detailed description of the trace from two models on the same task. This succinct analysis suggests that Claude exhibits a degree of robustness, as it is able to recover from past errors. Future work will focus on conducting a deeper analysis of Web Agent errors and developing automated methods for such evaluations.

**Table 2:** Results of a full round of experimentation on the unified benchmark ecosystem. The # Ep. column indicates the number of evaluation episodes per benchmark, which depends on the number of tasks and seeds used (See Appendix E for details). For budget reasons, grayed-out values (0.0) indicate skipped evaluations for the least-performing models on the hardest benchmark (WorkArena L3). The table also shows the total cost (USD) and average step per episode for the overall best-performing model, Claude 3.5 Sonnet. Costs for each LLM are given in Appendix F along with API cost details.

| Benchmark | # Ep. | Claude-3.5-Sonnet | GPT-4o | GPT-4o Mini | Lama-3.1 70B | Lama-3.1 405B | o1 Mini | Claude-3.5-Sonnet Cost ($) | Steps |
|---|---|---|---|---|---|---|---|---|---|
| MiniWoB | 625 | **69.8**$_{\pm 1.8}$ | 63.8$_{\pm 1.9}$ | 56.6$_{\pm 2.0}$ | 57.6$_{\pm 2.0}$ | 64.6$_{\pm 1.9}$ | 67.8$_{\pm 1.9}$ | 23.11 | 3.7 |
| WorkArena L1 | 330 | 56.4$_{\pm 2.7}$ | 45.5$_{\pm 2.7}$ | 27.0$_{\pm 2.4}$ | 27.9$_{\pm 2.5}$ | 43.3$_{\pm 2.7}$ | **56.7**$_{\pm 2.7}$ | 100.03 | 9.0 |
| WorkArena L2 | 235 | **39.1**$_{\pm 3.2}$ | 8.5$_{\pm 1.8}$ | 1.3$_{\pm 0.7}$ | 2.1$_{\pm 0.9}$ | 7.2$_{\pm 1.7}$ | 14.9$_{\pm 2.3}$ | 299.44 | 33.8 |
| WorkArena L3 | 235 | **0.4**$_{\pm 0.4}$ | 0.0$_{\pm 0.0}$ | 0.0$_{\pm 0.0}$ | 0.0$_{\pm 0.0}$ | 0.0$_{\pm 0.0}$ | 0.0$_{\pm 0.0}$ | 191.50 | 25.0 |
| WebLINX | 2,650 | **13.7**$_{\pm 0.6}$ | 12.5$_{\pm 0.6}$ | 11.6$_{\pm 0.6}$ | 8.9$_{\pm 0.5}$ | 7.9$_{\pm 0.5}$ | 12.5$_{\pm 0.6}$ | 104.62 | 1.0 |
| WebArena | 812 | **36.2**$_{\pm 1.7}$ | 31.4$_{\pm 1.6}$ | 17.4$_{\pm 1.3}$ | 18.4$_{\pm 1.4}$ | 24.0$_{\pm 1.5}$ | 28.6$_{\pm 1.6}$ | 138.76 | 6.8 |
| VisualWebArena | 910 | 21.0$_{\pm 1.3}$ | **26.7**$_{\pm 1.5}$ | 16.9$_{\pm 1.2}$ | - | - | - | 134.69 | 4.7 |
| AssistantBench | 181 | 5.2$_{\pm 1.5}$ | 4.8$_{\pm 2.4}$ | 2.1$_{\pm 1.0}$ | 2.8$_{\pm 1.1}$ | 3.9$_{\pm 1.0}$ | **6.9**$_{\pm 2.2}$ | 37.33 | 7.5 |

# 7 Discussion

In this work, we introduced the BrowserGym ecosystem, which contributes to the standardization of web agent research by providing a unified environment for evaluating web agents. We make our contribution available as two easy-to-use Python packages, BrowserGym and AgentLab. The first provides a unified platform exposing all existing web agent benchmarks under the same interface, and is designed to facilitate the addition of new benchmarks. AgentLab, on the other hand, provides a complementary interface and set of tools for the implementation and evaluation of web agents, with a set of flexible web agent implementations, an expandable LLM / VLM API, reproducibility features, trace analysis tools, as well as an online leaderboard.

To showcase our main contribution, we conducted the first large-scale empirical evaluation of a representative set of state-of-the-art LLM / VLM models on six popular web agent benchmarks, using AgentLab and BrowserGym. While running such an experiment using the original, fragmented benchmark codes would be extremely cumbersome, BrowserGym makes all of them accessible through the same interface and allows for large-scale, unified experiments in a single code base with AgentLab. By fostering reproducibility and facilitating consistent benchmarks, the BrowserGym ecosystem lays the groundwork for more dependable and insightful evaluations of web agent capabilities, thus accelerating the advancement of LLM-driven automation.

## 7.1 Limitations

**Reproducibility**   Despite our efforts to improve reproducibility, stochasticity, and external factors remain significant challenges. Localization differences, such as time zones, default languages, and geographic settings, can lead to inconsistent agent behaviors when interacting with live websites. Additionally, variations in operating systems and browsers can affect how web pages are rendered, impacting the uniformity of results. The presence of dynamic elements, such as advertisements, adds to non-determinism, leading agents to face different environments during repeated tasks, which affects their consistency. Addressing these challenges is essential for reliable and repeatable benchmarks in web agent research.

**Risks**   Letting an autonomous agent act on the world-wide-web on a human's behalf can have consequences, and prove to be problematic on benchmarks that require an open-web access (AssistantBench, soon GAIA). This is currently mitigated in BrowserGym via URL protection (MiniWoB, WorkArena, WebArena / VisualWebArena), plus a warning notice on BrowserGym and AgentLab's landing pages.

**Robot detection**   Robot detection mechanisms significantly limit web agents, especially for open-web tasks. Websites use techniques like CAPTCHA, IP rate limiting, and behavior analysis to block automated agents, preventing them from completing tasks effectively. This limits the ability of agents to interact freely with web environments and reduces the scope of tasks that can be automated. AssistantBench is the only benchmark impacted by those issues so far in BrowserGym, as it uses the open web as its information retrieval tool.

**Agent collisions**   The widespread use of databases on the web means that many tasks involve updating records and modifying entries. When multiple agents operate simultaneously, they can interfere with each other, especially when executing tasks that alter the same database entries. For example, if two agents add an item to their cart concurrently, they may unintentionally update each other's cart, leading to inconsistent outcomes. This issue is particularly problematic in environments like WebArena and VisualWebArena, where a single virtual machine is used for each agent. As a result, only one agent can be run at a time on these benchmarks, significantly slowing down the evaluation process.

**Synchronous interactions**   Another limitation is the sync loop mechanism used to manage interactions within the browser environment. For example, when agents need to perform multiple actions in quick succession, the sync loop may not keep up, leading to delays and reduced efficiency. This can result in performance bottlenecks, especially for complex tasks requiring rapid, sequential actions.

### 7.2 Avenue for Future Work

To further develop the field, several areas require attention. One promising direction is improving **safety** mechanisms for web agents. Currently, safety evaluation in web agent benchmarks, including ensuring agents' compliance with data privacy and safety policies, as well as their resilience to malicious interactions, is underexplored. The development of secure environments is crucial for adopting web agents in enterprise or sensitive applications.

Another avenue for future work involves the creation of real-time web agents capable of interacting with live environments at speeds comparable to human users. This necessitates improvements in both latency reduction and real-time decision-making, which would enhance the applicability of web agents in dynamic settings, such as customer service or rapid data retrieval.

The potential for smaller, yet efficient LLMs is also an important focus. Developing smaller models that retain the ability to understand and reason through complex web interactions would lower computational demands, making web automation more accessible and sustainable. This aligns with efforts to increase the efficiency of LLM-based agents, particularly for real-world deployment where computational resources may be constrained.

In addition, enhancing reactivity through the use of computer-level vision-language models (VLMs) can provide agents with a more nuanced understanding of the visual context on web pages. Current models often lack the capability to fully interpret complex visual layouts or graphical information, limiting their interaction fidelity with certain websites. Integrated models like Anthropic's computer use model could make web agents more versatile and capable of handling tasks that require both visual reasoning and textual inputs.

Another direction is to leverage inference time scaling, optimizing it specifically for web tasks and web agents. This could involve making better use of the model's self-reflection capabilities to further improve task performance, adapting dynamically to various complexities encountered during web navigation, and efficiently managing resources to reduce latency. By tailoring these improvements, web agents can become more effective at real-time, adaptive decision-making, ultimately enhancing user experiences and broadening their applicability.

As BrowserGym saves and logs every piece of information seen in its experiments, it could serve as a valuable collection tool for finetuning models. This collected data could be utilized to further refine agent behaviors, improve model performance on specific web tasks, and create efficient agents capable of handling complex scenarios more efficiently.

## 8 Broader Impact

The development of more general UI-based agents holds the potential to significantly transform how work is performed on computers. These agents promise to automate tasks at an unprecedented pace, with reduced costs, and with a broader and deeper scope of expertise. While current benchmarks demonstrate that this level of sophistication has not yet been achieved, the rapid pace of advancements in the field suggests that such a future may not be far off. This progress could unlock immense economic value but is also likely to result in substantial and rapid job displacement, presenting a critical societal challenge.

AI safety must be central to this emerging agentic revolution. Presently, jailbreaking a large language model (LLM) might only expose knowledge already available on the web. However, as agents become more sophisticated, vulnerabilities such as prompt-injection attacks could be exploited to leak sensitive company information, authorize unwanted transactions, or cause other unintended harm. The widespread deployment of agents at scale will create enormous financial incentives for malicious actors to develop targeted attacks that uncover and exploit weaknesses in these systems.

The rise of UI-based agents also introduces new challenges and considerations regarding digital advertising. As these agents navigate the web autonomously, their interactions with dynamic ad content could disrupt existing advertising models, impacting both advertisers and publishers. Unlike human users, agents are unlikely to engage with ads in traditional ways, which may lead to shifts in how ad impressions, clicks, and conversions are measured and monetized. Additionally, the ability of agents to filter, block, or bypass advertisements could create conflicts with current web monetization strategies. To maintain a fair and functional digital ecosystem, actors from both sides should explore standardized approaches for handling ad-related content in a way that balances automation efficiency with the sustainability of online advertisement business models.

As these technologies evolve, society must proactively address the accompanying challenges by establishing robust policies and frameworks. Rapid action will be essential to ensure that the benefits of these advancements are maximized while minimizing potential risks and harm.

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

# A  BrowserGym's High-Level Action Set

**Table 3:** Complete list of high-level action set primitives available when using BrowserGym's `HighLevelActionSet` feature.

| Category | Primitive | Description |
|---|---|---|
| bid | `fill(bid, text)` | Fill an input field with text. |
| | `click(bid, button)` | Click an element. |
| | `dblclick(bid, button)` | Double-click an element. |
| | `hover(bid)` | Hover the mouse over an element. |
| | `press(bid, key_comb)` | Focus an element and press a combination of keys. |
| | `focus(bid)` | Focus an element. |
| | `clear(bid)` | Clear an input field. |
| | `select_option(bid, options)` | Select one or multiple options in a drop-down element. |
| | `drag_and_drop(from_bid, to_bid)` | Drag and drop one element to another. |
| | `upload_file(bid, file)` | Click a 'filechooser' element, then select one or multiple input files for upload. |
| coord | `mouse_move(x, y)` | Move the mouse to a location. |
| | `mouse_down(x, y, button)` | Move the mouse then press and hold a button. |
| | `mouse_up(x, y, button)` | Move the mouse then release a button. |
| | `mouse_click(x, y, button)` | Move the mouse and click a button. |
| | `mouse_dblclick(x, y, button)` | Move the mouse and double-click a button. |
| | `mouse_drag_and_drop(from_x, from_y, to_x, to_y)` | Drag and drop from a location to a location. |
| | `mouse_upload_file(x, y, file)` | Click a 'filechooser' location, then select one or multiple input files for upload. |
| | `keyboard_down(key)` | Press and holds a keyboard key. |
| | `keyboard_up(key)` | Release a keyboard key. |
| | `keyboard_press(key_comb)` | Press a combination of keys. |
| | `keyboard_type(text)` | Types a string of text through the keyboard. |
| | `keyboard_insert_text(text)` | Insert a string of text in the currently focused element. |
| tab | `new_tab()` | Open a new tab. |
| | `tab_close()` | Close the current tab. |
| | `tab_focus(index)` | Bring a tab to front (activate tab). |
| nav | `go_back()` | Navigate to the previous page in history. |
| | `go_forward()` | Navigate to the next page in history. |
| | `goto(url)` | Navigate to a url. |
| misc | `send_msg_to_user(message)` | Send a message to the user in the chat. |
| | `report_infeasible(reason)` | Send a special message in the chat and terminate. |
| | `scroll(dx, dy)` | Scroll pixels in X and/or Y direction. |
| | `noop(seconds)` | Wait and do nothing. |

```
20 different types of actions are available.

click(bid: str, button: Literal['left', 'middle', 'right'] = 'left', modifiers: list = [])
    Description: Click an element.
    Examples:
        click('a51')
        click('b22', button='right')
        click('48', button='middle', modifiers=['Shift'])

fill(bid: str, value: str)
    Description: Fill out a form field. It focuses the element and triggers an input event with the entered text. It works for
    ↪  <input>, <textarea> and [contenteditable] elements.
    Examples:
        fill('237', 'example value')
        fill('45', 'multi-line\nexample')
        fill('a12', 'example with "quotes"')

scroll(delta_x: float, delta_y: float)
    Description: Scroll horizontally and vertically. Amounts in pixels, positive for right or down scrolling, negative for left or
    ↪   up scrolling. Dispatches a wheel event.
    Examples:
        scroll(0, 200)
        scroll(-50.2, -100.5)

send_msg_to_user(text: str)
    Description: Sends a message to the user.
    Examples:
        send_msg_to_user('Based on the results of my search, the city was built in 1751.')

report_infeasible(reason: str)
    Description: Notifies the user that their instructions are infeasible.
    Examples:
        report_infeasible('I cannot follow these instructions because there is no email field in this form.')

...

Only a single action can be provided at once. Example:
fill('a12', 'example with "quotes"')
```

**Figure 10:** Example description of a high-level action set in BrowserGym (`HighLevelActionSet.describe()`).

# B   Detailed Benchmark Descriptions

**MiniWoB** [15] (Shi et al., 2017; Liu et al., 2018) Each task only requires accessing a single HTML page, the entire logic of the task and its validation is done in JavaScript. As such, MiniWoB natively supports parallelization since there are no collisions by design.

**WebArena** [16] (Zhou et al., 2024b) Contains 6 domains mimicking real-world websites, self-hosted via Docker containers. Docker containers must be reset to their initial state before each new agent evaluation. Task validation is performed via logic rules (HTML content, URL match, message content), using either exact matching or semantic matching using a GPT-3.5 model as a judge (hence implies some costs).

**VisualWebArena** [17] (Koh et al., 2024a) Based on the WebArena implementation, VisualWebArena retains 2 of the original domains, plus includes a new one. Task goals include images for vision-based tasks, such as "Find a pair of shoes that looks like this image". Docker containers must be reset to their initial state before each new agent evaluation, and some tasks require an explicit reset of a specific Docker. Task validation is similar to WebArena, and also relies on visual matching using an open-source VLM (`blip2-flan-t5-xl`).

**WorkArena** [18] (Drouin et al., 2024; Boisvert et al., 2024) The backend platform requires a Personal Developer Instance (PDI), available on demand with a free ServiceNow user account. Validation is done via logic rules, using database queries to check for specific content in the backend, and JavaScript code injected on the platform's pages to check for specific actions in the frontend. WorkArena natively supports large-scale parallel agent evaluation, as it prevents trajectory collisions by design.

**AssistantBench** [19] (Yoran et al., 2024) Contains 214 realistic and time-consuming open-domain tasks. Tasks require browsing more than 525 web pages from 258 different websites to answer correctly. The dataset covers a diverse range of topics and is composed of a sub-set of general assistant tasks and a sub-set of domain-specific tasks created by domain experts. The benchmark features a small development set, consisting of 33 tasks and a larger test set of 181 tasks. The gold answers are hidden for the test, but released for the development set in addition to the URLs from which the answers can be inferred and and answering strategies. Additional task-level metadata includes difficulty and time-dependency, indicating whether the answer is static or includes stable information that could change in the future, e.g., a business's opening hours. Since the test set is hidden, predictions need to be submitted for evaluation. This is supported via an API to enable easy evaluations at scale.

**WebLINX** [20] (Lù et al., 2024) Static, supervised benchmark. Interaction traces in the original dataset are converted to the BrowserGym observation/action format, and each task consists of predicting an action that must match a recorded action taken by a human during a demonstration. As a result, it differs from environment-based benchmarks by ending after one step, while formulating each recorded action as a different task. Consequently, validation produces scalar rewards via partial matching, using the original metrics selected by Lù et al. (2024).

---

[15]https://miniwob.farama.org/

[16]https://webarena.dev/

[17]https://github.com/web-arena-x/visualwebarena

[18]https://servicenow.github.io/WorkArena/

[19]https://assistantbench.github.io/

[20]https://mcgill-nlp.github.io/weblinx/

## C  Example Prompt

In Figure 11, we show the basic usage of AgentLab's dynamic prompting tools.

```python
from agentlab.agents import dynamic_prompting as dp

class ExampleAgent(Agent):
    def __init__(self, flags, chat_model_args):
        self.flags = flags
        self.action_set = HighLevelActionSet(["bid"])

        # unified LLM/VLM API for using different provider
        # e.g.: OpenAI, OpenRouter, Azure, or self hosted using TGI.
        self.chat_llm = chat_model_args.make_model()

    def get_action(self, obs):
        instructions = dp.GoalInstructions(obs["goal_object"])
        observation = dp.Observation(obs, self.flags.obs)
        action = dp.ActionPrompt(self.action_set, action_flags=self.flags.action)

        prompt = "\n".join([
            instructions.prompt, # the goal of the task
            observation.prompt, # html and/or ax_tree depending on flags.obs
            action.prompt, # the action space described based on flags.action
            action.concrete_ex, # concrete example of the expected answer
        ])

        # Progressively shrink some part of the prompt until it fits the limit
        # Each component has its own shrinking strategy
        prompt = dp.fit_tokens(prompt, max_prompt_tokens=40000)

        action_str = action.parse_answer(self.chat_llm(prompt))
        return action_str, bgym.AgentInfo(chat_messages=[prompt])

    def obs_preprocessor(self, obs):
        obs["processed_html"] = process_html(obs["dom_object"])
        return obs
```

```python
class ObsFlags:
    use_html: bool
    use_ax_tree: bool
    use_tabs: bool
    use_focused_element: bool
    use_error_logs: bool
    use_history: bool
    use_past_error_logs: bool
    use_action_history: bool
    use_screenshot: bool
    use_som: bool
    extract_visible_tag: bool
    extract_clickable_tag: bool
    extract_coords: bool
    filter_visible_elements_only: bool
    filter_with_bid_only: bool
    filter_som_only: bool

class ActionFlags(Flags):
    action_set:
    ↪  bgym.HighLevelActionSetArgs
    long_description: bool
    individual_examples: bool
```

**Figure 11:** Example of how to implement an agent using dynamic prompting and the unified LLM API. For simplicity, only a subset of the prompt components and flags are depicted. Each prompt component has its own shrinking mechanism which allows for reducing the size of the prompt dynamically when it exceeds the maximum number of allowed tokens.

Concretely, here is an example prompt from `GenericAgent` on WorkArena L2, at step 2:

**GenericAgent example prompt**

```
# Instructions
Review the current state of the page and all other information to find the best
possible next action to accomplish your goal. Your answer will be interpreted
and executed by a program, make sure to follow the formatting instructions.

## Goal:

Managing Your Existing Expenses

 Concretely, you need to complete the following steps:
1. Navigate to the "Expense Lines" module of the "Cost" application.
2. Create a filter for the list to extract all entries where:
    - "Short description" contains "#SERIES-d1e0dc99-5"
3. Delete expense lines with duplicated short descriptions, keeping only the most expensive duplicate.

# Observation of current step:

## Currently open tabs:
Tab 0 (active tab):
    Title: Home | ServiceNow
    URL: https://dev255323.service-now.com/now/nav/ui/home

## AXTree:
Note: [bid] is the unique alpha-numeric identifier at the beginning of lines for each element in the AXTree. Always use bid to refer to elements in your actions.

Note: You can only interact with visible elements. If the "visible" tag is not
present, the element is not visible on the page.

RootWebArea 'Classic | Unified Navigation App | ServiceNow'
        [47] generic, live='assertive', atomic, relevant='additions text'
        [48] generic, live='polite', atomic, relevant='additions text'

...

# History of interaction with the task:
```

## step 0

<think>
Looking at the current page, I need to navigate to the "Expense Lines" module in the "Cost" application. First, I need to access the "All" menu since it's likely where I'll find the Cost application. The "All" menu button is visible with bid "79".
</think>

<action>
click('79')
</action>

## step 1

<think>
I see that the "All" menu is already expanded. I need to find the "Cost" application in the list of applications. Looking through the AXTree, I can see various applications listed, but I need to search for "Cost" since it's not immediately visible. I can use the search box at the top of the All menu which has bid "242" and is currently focused.
</think>

<action>
fill('242', 'Cost')
</action>

## step 2

<think>
I can see that the search box is already filled with "Cost" and the menu is showing filtered results. I can see the "Expense Lines" link with bid "1145" which is exactly what I need to navigate to. I'll click on this link to access the Expense Lines module.
</think>

<action>
click('1145')
</action>

# Action space:
Note: This action set allows you to interact with your environment. Most of them
are python function executing playwright code. The primary way of referring to
elements in the page is through bid which are specified in your observations.

20 different types of actions are available.

```
noop(wait_ms: float = 1000)
send_msg_to_user(text: str)
report_infeasible(reason: str)
scroll(delta_x: float, delta_y: float)
fill(bid: str, value: str)
select_option(bid: str, options: str | list[str])
click(bid: str, button: Literal['left', 'middle', 'right'] = 'left', modifiers: list[typing.Literal['Alt', 'Control', 'ControlOrMeta', 'Meta', 'Shift']] = [])
dblclick(bid: str, button: Literal['left', 'middle', 'right'] = 'left', modifiers: list[typing.Literal['Alt', 'Control', 'ControlOrMeta', 'Meta', 'Shift']] = [])
hover(bid: str)
press(bid: str, key_comb: str)
focus(bid: str)
clear(bid: str)
drag_and_drop(from_bid: str, to_bid: str)
upload_file(bid: str, file: str | list[str])
tab_close()
tab_focus(index: int)
new_tab()
go_back()
go_forward()
goto(url: str)
```
Only a single action can be provided at once. Example:
fill('a12', 'example with "quotes"')

Note:
* Some tasks may be game like and may require to interact with the mouse position
in x, y coordinates.
* Some text field might have auto completion. To see it, you have to type a few
characters and wait until next step.
* If you have to cut and paste, don't forget to select the text first.
* Coordinate inside an SVG are relative to it's top left corner.
* Make sure to use bid to identify elements when using commands.
* Interacting with combobox, dropdowns and auto-complete fields can be tricky,
sometimes you need to use select_option, while other times you need to use fill
or click and wait for the reaction of the page.

# Abstract Example

Here is an abstract version of the answer with description of the content of
each tag. Make sure you follow this structure, but replace the content with your
answer:

<think>
Think step by step. If you need to make calculations such as coordinates, write them here. Describe the effect
that your previous action had on the current content of the page.
</think>

<action>
One single action to be executed. You can only use one action at a time.
</action>

# Concrete Example

Here is a concrete example of how to format your answer.
Make sure to follow the template with proper tags:

<think>
From previous action I tried to set the value of year to "2022",
using select_option, but it doesn't appear to be in the form. It may be a
dynamic dropdown, I will try using click with the bid "a324" and look at the
response from the page.
</think>

```
</think>

<action>
click('a324')
</action>
```

# D  GenericAgent in depth

**Table 4:** List of `GenericAgent`'s prompting options. The setting column gives the flags used for our experimentations, with the exception of the agents ran on VisualWebArena that have `use_screenshot=True`

| Category | Flag | Setting | Description |
|---|---|---|---|
| | `use_html` | ✗ | Adds the HTML content to the prompt. |
| | `use_axtree` | ✓ | Adds the AXTree content to the prompt. |
| | `use_focused_element` | ✓ | Specifies which element is focused at the moment. |
| | `use_error_logs` | ✓ | Show the previous action error (e.g. action not possible, timed out...). |
| | `use_past_error_logs` | ✗ | Show all previous errors. |
| | `use_action_history` | ✓ | Display a list of all previous actions. |
| | `use_think_history` | ✓ | Display a list of all previous Chain-of-thought reasonings. |
| observation | `use_screenshot` | ✗ | Switch to vision mode. |
| | `use_som` | ✗ | Agent uses Set-of-Marks instead of screenshots. |
| | `extract_visible_tag` | ✓ | For each HTML/AXTree element, add a tag that if it is visible. |
| | `extract_clickable_tag` | ✓ | For each HTML/AXTree element, add a tag that if it is clickable. |
| | `extract_coords` | ✗ | For each HTML/AXTree element, add its coordinates. |
| | `filter_visible_elements_only` | ✗ | Only show int the HTML/AXTree elements that are visible. |
| | `long_description` | ✗ | Adds the full docstring of each action. |
| action | `individual_examples` | ✗ | Adds a usage example for each action function. |
| | `multi_actions` | ✗ | Allows performing multiple actions in one step. |
| | `use_thinking` | ✓ | Use Chain-of-thoughts reasoning. |
| | `use_plan` | ✗ | The agents provides and refines a plan at each step. |
| | `use_criticize` | ✗ | The LLM must draft, then criticize an action before outputing the real action. |
| reasoning | `use_concrete_example` | ✓ | Whether or not to use a real example for in-context learning. |
| | `use_abstract_example` | ✓ | Whether or not to use an abstract example with descriptive instructions. |
| | `extra_instructions` | N/A | A string to be passed to the prompt as an extra goal. |

# E  Benchmark setting

In Table 5 we give some more information about the benchmark settings during our experiments. For Miniwob and WorkArena L1, we use respectively 5 and 10 seeds per task. WebArena and its visual counterpart are meant to be used as a whole, as they do not provide task seeding. WorkArena L2 and L3 are shipped with preset curricula to balance out the types of tasks in an experiment. Finally, AssistantBench and WebLINX provide test sets, which are used to evaluate the models.

**Table 5:** Summary of experiment settings for each benchmark.

| Benchmark | # Tasks | # Seeds | Max Steps | Resulting # runs |
|---|---|---|---|---|
| MiniWoB | 125 | 5 | 10 | 625 |
| WebArena | 812 | / | 30 | 812 |
| VisualWebArena | 910 | / | 30 | 910 |
| WorkArena L1 | 33 | 10 | 30 | 330 |
| WorkArena L2 | 341 | curriculum | 50 | 235 |
| WorkArena L3 | 341 | curriculum | 50 | 235 |
| WebLINX | 31586 | test split | 1 | 2650 |
| Assistant Bench | 214 | / | 30 | 181 |

## F   Costs and API Usage

In Table 6 we provide more detailed data on the cost of our experiments. We also show token counts. We highlight that the Llama models were ran using the OpenRouter API, which relies on different providers to give users access to those models. Prices for those two models may have changed since running our experiments.

| Agent | Total Cost ($) | Input Tokens (M tokens) | Output Tokens (M tokens) | API Input Cost ($/M tokens) | API Output Cost ($/M tokens) |
|---|---|---|---|---|---|
| Claude | 894.80 | 276.20 | 4.41 | 3.00 | 15.00 |
| GPT-4o | 720.72 | 277.25 | 2.76 | 2.50 | 10.00 |
| GPT-4o-mini | 75.73 | 479.72 | 6.29 | 0.15 | 0.60 |
| Llama-3.1 405B | 849.63 | 359.89 | 8.13 | 2.30 | 2.30 |
| Llama-3.1 70B | 78.79 | 221.23 | 5.02 | 0.35 | 0.35 |
| o1 Mini | 971.12 | 171.25 | 38.11 | 3.00 | 12.00 |

**Table 6:** Comparison of experiment costs between models. Values are summed over all benchmarks, except VisualWebArena.

## G   Experiment runtime and hardware

We provide detailed information on total runtimes and benchmark durations (Table 7). These metrics are presented for our best-performing model, Claude 3.5 Sonnet. For each benchmark, we report the following key metrics:

- **Cumulative Experiment Duration**: The overall time taken for the benchmark experiment to complete. This value does not account for parallelization.

- **Average Step Duration**: The mean time taken per step in the experiment.

- **Cumulative Environment Runtime**: The total time spent running the environment, excluding agent processing time.

- **Average Environment Runtime per Step**: The average time spent by the environment per step.

- **Cumulative Number of Environment Steps**: The total count of steps executed within the environment during the experiment.

Discrepancies between the total runtime and environment runtime can be observed, primarily due to the API used. Slower or overcrowded APIs may introduce significant variations.

It is important to note that the reported durations do not account for parallelization. In practice, experiments were conducted using up to 20 parallel jobs depending on the benchmark. This means that while the reported total experiment duration reflects the absolute runtime of the benchmark, the actual time required to complete an experiment may be significantly lower due to parallel execution. The level of parallelization was adjusted based on the computational requirements of each benchmark. Lighter benchmarks, such as MiniWoB, could support up to 20 parallel runs without issue, whereas more resource-intensive benchmarks, like WebArena or WorkArena, required limiting parallel jobs to prevent server overload and ensure stable execution.

Our experiments were conducted on large-scale compute clusters equipped with Intel(R) Xeon(R) Gold 6126 CPUs @ 2.60GHz and effectively unlimited RAM. Given the relatively low computational demands of our benchmarks, experiments could also be run locally on standard laptops without significant performance degradation. However, the primary constraints arise from the benchmark environments themselves. For example, WebArena experiments were executed on Azure VMs with 8 CPUs and 32GB RAM, which imposed limitations on execution speed and parallelization capabilities.

## H   Results analysis

In this section, we compare the execution of a task by our `GenericAgent`, backed by two different LLMs, Claude 3.5 Sonnet which executed the task correctly, and GPT-4o which failed the task. We will be looking at a task from the very challenging WorkArena L2, on which Claude achieves groundbreaking performances.

| Benchmark | Cumulative Experiment Duration (hours) | Avg. Step Duration (seconds) | Cumulative Environment Duration (hours) | Avg. Environment Step Duration (seconds) | Cumulative Steps |
|---|---|---|---|---|---|
| AssistantBench | 4.0 | 9.1 | 2.0 | 4.5 | 1590 |
| MiniWoB | 3.1 | 5.0 | 0.7 | 1.1 | 2282 |
| WebArena | 18.6 | 12.2 | 11.6 | 7.6 | 5493 |
| WebLINX | 4.5 | 6.1 | 0.1 | 0.2 | 2649 |
| WorkArena | 8.2 | 9.9 | 4.4 | 5.3 | 2985 |
| WorkArena L2 | 23.6 | 10.7 | 10.9 | 4.9 | 7947 |
| WorkArena L3 | 16.1 | 9.8 | 7.4 | 4.5 | 5884 |
| VisualWebArena | 16.9 | 14.1 | 10.5 | 8.8 | 4319 |

**Table 7:** Summary of runtime metrics for Claude 3.5 Sonnet across various benchmarks. Cumulative durations indicate total execution time without accounting for parallelization. Environment-related durations reflect only the time spent in the environment, excluding agent processing overhead.

We focus on the task `workarena.servicenow.navigate-and-order-standard-laptop-l2`, which has the following goal:

```
Order a standard laptop from the service catalog

 Concretely, you need to complete the following steps:
1. Navigate to the "Service Catalog" module of the "Self-Service" application.
2. Go to the hardware store and order 6 "Standard Laptop" with configuration {'Additional software requirements': 'Salesforce,
↪  Microsoft Office 365, Asana, HubSpot', 'Adobe Acrobat': True, 'Adobe Photoshop': True}
```

Figure 12 displays the trajectory taken by Claude for this task. Interestingly, in step 12e the agent manages to overcome an environment limitation. Clicking on one of the element of the checkbox does not trigger the checkbox, and the agent gives the following reasoning before correctly checking the checkbox:

```
I see that clicking directly on the checkbox didn't work because the label intercepts pointer events. Looking at the AXTree, I can
↪  see that the label for Adobe Acrobat has bid "a184". I'll try clicking on the label instead, which should toggle the checkbox.
```

The agent then proceeds to complete the task.

On the other hand, Figure 13 displays the last two steps from the same task, executed by GPT-4o. The agent does not notice it is only ordering one laptop instead of the required 6, failing the task.

More precisely, the agent attempts to click on element `a243` instead of using the `select_option` action in the following AXTree:

```
[a237] combobox 'Quantity' value='1', visible, hasPopup='menu', expanded=False, controls='quantity_label_span'
    [a238] menuitem '1', selected=True
    [a239] menuitem '2', selected=False
    [a240] menuitem '3', selected=False
    [a241] menuitem '4', selected=False
    [a242] menuitem '5', selected=False
    [a243] menuitem '6', selected=False
    [a244] menuitem '7', selected=False
    [a245] menuitem '8', selected=False
    [a246] menuitem '9', selected=False
    [a247] menuitem '10', selected=False
```

This causes the action to fail. However, seeing the previous Chain-of-thought reasoning in its prompt, GPT-4o continues with the following:

```
I need to proceed with configuring the "Standard Laptop" order. The quantity has been set to 6. Now, I should check the boxes for
↪  "Adobe Acrobat" and "Adobe Photoshop" to include them in the order.
```

The agent fools itself and does not understand that the previous action was in fact a failure. It then goes on to complete the task without correcting this mistake.

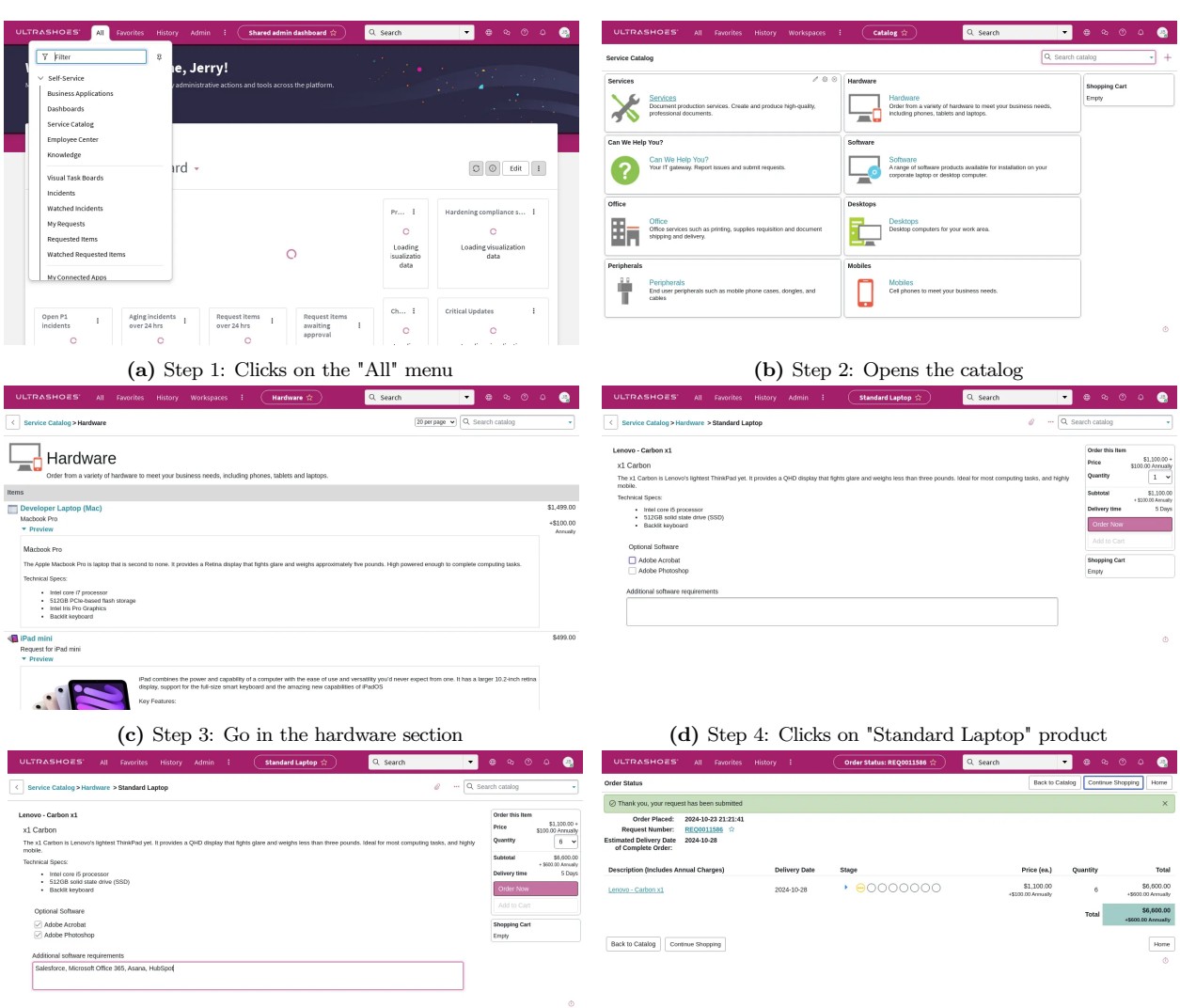

(a) Step 1: Clicks on the "All" menu

(b) Step 2: Opens the catalog

(c) Step 3: Go in the hardware section

(d) Step 4: Clicks on "Standard Laptop" product

(e) Step 5: Fill in the required fields (multiple steps)

(f) Step 6: Complete order

**Figure 12:** Successful execution of the task `workarena.servicenow.navigate-and-order-standard-laptop-l2`, seed 99, by Claude 3.5 Sonnet. The Steps given in the caption are not the same as the actual task steps, for the sake of conciseness.

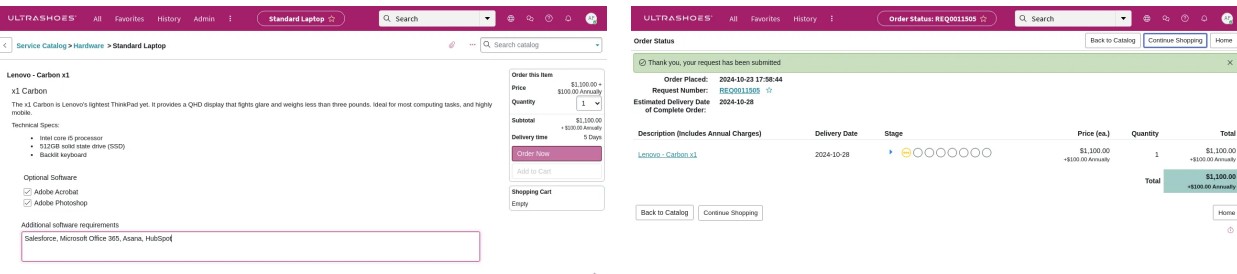

**(a)** The agent unsuccessfully attempts to set the amount to 6

**(b)** Considering the order correct, the agent completes a wrong order.

**Figure 13:** Failed steps during the execution of the task `workarena.servicenow.navigate-and-order-standard-laptop-l2`, seed 99, by GPT-4o

