# OpenReview forum: "The BrowserGym Ecosystem for Web Agent Research"
_TMLR — Accepted by TMLR_

### Review · Reviewer_dn9K · 2024-12-23

**Summary Of Contributions:**

The paper introduces a comprehensive ecosystem designed to standardize and streamline web agent research. The main contribution is BrowserGym, a unified framework that enables researchers to evaluate and compare different web agents across multiple benchmarks using a consistent interface and methodology. In addition, the AgentLab framework provides tools for creating, testing, analyzing, and managing experiments, including features for parallel execution and reproducibility monitoring. The authors demonstrate the utility of the ecosystem by conducting a comparative study of six state-of-the-art language models on different web agent benchmarks.

**Audience:**

Yes

**Broader Impact Concerns:**

This work raises an interesting consideration regarding web agents' interaction with dynamic advertising content. Although the authors appropriately address the reproducibility challenges associated with such content, future ecosystem development could benefit from considering how to handle advertisement-related content in a standardized manner. Currently, network-related information (normally inspected against blacklists to identify and block ad sources) is not included in agent observations, and while this may not have a significant impact on current live benchmarks, it could become relevant as web agents become more sophisticated and spreaded. As the ecosystem aims to serve as a standard for the evaluation of web agents, a consideration of how to handle such content could help prevent potential future challenges in the interaction between web agents and dynamic advertising systems.

**Claims And Evidence:**

Yes

**Requested Changes:**

Critical changes:
- Avoid unnecessary code implementation details and function/class nomenclature as they reduce readability and can be easily avoided (especially in Sections 3.2, 5.1, 5.5, 5.6).
- Instead, where code and variable/function/class names are considered necessary or more effective than a textual description, be sure to:
    - Avoid the use of `var_name` as a name in the sentence, e.g. Section 3.3 "Starting from a blank `page`" -to-> "Starting from a blank `page` object"  or "Starting from a blank page"
    - Follow consistent nomenclature throughout the text: `ClassName`, `variable_name`, `function_name()` seems to be the standard adopted, but not adhered to everywhere (e.g. `action_mapping` is used as both function and object, `Study` object is capitalized but not everywhere, ...)
- Section 5.2: "[...] the actual compute requirement of the job is _relatively small_ and we can launch up to 20 tasks in parallel on a _single laptop_ or even 50-100 on a server machine _with many CPU cores and large memory_" is too general. Replace vague descriptions of hardware and, if possible, provide some metrics to quantify computing time/resources.
- Section 5.4.2 - Reproducible Journal: Provide a clearer explanation of the feature and its role.
- Section 6.2: Is it correct to say that the Llama models and o1 Mini have an unfair disadvantage in the comparison presented, because the other methods are vision-enabled and are also fed the web page screenshot? If so, this should be emphasized more in the text.
- Section 6.2: The last paragraph states that: "This result suggests that the BrowserGym API for browser interaction is more adapted for action-oriented tasks than for pure question-answering tasks.", but it is not clear what problem/limitation is causing the low performance and thus this conclusion.

Minor changes:
- Missing citation to [Gymnasium](https://arxiv.org/abs/2407.17032) framework.
- Section 3.1 - Extra element properties: Define and explain the SoM indicator and its purpose.
- Section 5.4.2 - Reproduced results in the leaderboard: Clarify what qualifies an agent as "reproducible" and better explain the leaderboard's role (up to this point is only barely mentioned in the contributions of Section 1 therefore it should be explained what is it).
- Section 5.4.2 - ReproducibilityAgent: Better define what the ReproducibilityAgent is (function? class?).
- Section 6.2: Similarly to the request for Section 5.2, if possible, it would be helpful to have metrics that relate performance to time.
- Table 2: The results table includes a cost column that, from what I can conclude, refers to the monetary cost of the model's API for evaluation. This is not exactly clear, plus I don't see the value added in reporting this information (neither the currency nor the cost per token of the API is specified, so this only provides a sense of scale and not much else).

Non-essential changes (minor suggestions on style and form):
- Section 2.2: "Due to their immense potential, there has been a substantial effort in developing web agents, _and_ agents differ in how they interact with the web environment, the backbone models, and the goal they aim to achieve, among others.", the _and_ is improperly used as a conjunction of two unrelated clauses.
- Section 2.2: "[...] enhance agent reasonning capabilities" -to-> "... reasoning ...".
- Section 2.2: "To address those important points, we developed BrowserGym and Agentlab, which we describe in detail in the subsequent sections." -to-> "... AgentLab ...".
- Section 3.1 - Screenshot: Add missing references to Figure 4.a and 4.b.
- Section 3.1 - Open tabs: "In BrowserGym, both _agents_ and _tasks_ have the option to use multiple tabs if needed". Distinguish between agent _choice_ and task _requirement_ for multiple tabs.
- Section 5: Consider moving agent creation description to beginning of section.
- Figure 9: `AgentXRay` -to-> AgentXRay (for consistency with other occurrences in the manuscript).
- Section 5.4.1 - Live websites: Restructure the discussion to focus on general "core factors that can lead to discrepancies in web agent evaluation" and not on the characteristics of the individual benchmark, to improve fluidity and consistency with the other paragraphs.
- Sec 5.4.1: In the first paragraph "Software version" is used as title while in the last paragraph "Non-deterministic tasks" is referenced as being part of the sentence.
- Section 5.5: Add missing leading space in first comment of `MyAgent` class figure.
- Sec 6.1: The first paragraph describes the details of two configurations: agent and observation. From what I can conclude the two configurations are very distinct; if not, the description is misleading.
- Sec 6.2 "Llama-3.1-405B surpasses GPT-4o-mini"  -> "... GPT-4o-Mini"

**Strengths And Weaknesses:**

Strengths:
- The paper presents comprehensive documentation of ecosystem features, providing details on customization and demonstrating minimum configuration requirements for evaluation purposes.
- The framework demonstrates considerable flexibility due to its modular architecture, specifically addressing key aspects:
  1. Forward compatibility with emerging evaluation tasks
  2. Customization of agent configuration
  3. Extensibility for integration of new LLMs and benchmarks.
  4. Efforts in relation to reproducibility concerns
- The web agent evaluation presented encompasses a number of different models, including proprietary, open-source, and “reasoning”-focused variants.

Weaknesses:
- The manuscript raises significant concerns about its eligibility under TMLR [Guidelines and Editorial Policies](https://jmlr.org/tmlr/editorial-policies.html#:~:text=In%20particular%2C%20we%20do%20not%20accept%20submissions%20that%20are%20expanded%20versions%20of%20conference%20papers.) "TMLR only accepts original contributions that don’t reuse the authors’ own prior work. In particular, we do not accept submissions that are expanded versions of conference papers.". There appears to be substantial content overlap with an earlier work [Drouin et al., ICML 2024] which also introduces the BrowserGym ecosystem.
- The manuscript often emphasizes implementation-specific details and code structure at the expense of higher-level feature descriptions. While the technical details are valuable, this approach causes the focus to shift from the broader conceptual framework and core utilities to implementation details. A more balanced presentation that prioritizes functional descriptions over implementation specifics would make the text more accessible to a wider audience.

---

> ### Author Response · Authors · 2025-02-02
> **Reviewer-Specific Answer**
>
> ## Corrections and style updates
>
> >Instead, where code and variable/function/class names are considered necessary or more effective than a textual description, be sure to:
> >	- Avoid the use of var_name as a name in the sentence, e.g. Section 3.3 "Starting from a blank page" -to-> "Starting from a blank page object" or "Starting from a blank page"
> >	- Follow consistent nomenclature throughout the text: ClassName, variable_name, function_name() seems to be the standard adopted, but not adhered to everywhere (e.g. action_mapping is used as both function and object, Study object is capitalized but not everywhere, ...)
>
> We updated the code usage in our paper to follow this remark. We also updated section 5's tone to fit with the other sections.
>
> >Section 5.4.2 - Reproducible Journal: Provide a clearer explanation of the feature and its role.
>
> The paragraph was updated to describe more clearly our Reproducibility Journal.
>
> >Section 6.2: Is it correct to say that the Llama models and o1 Mini have an unfair disadvantage in the comparison presented, because the other methods are vision-enabled and are also fed the web page screenshot? If so, this should be emphasized more in the text.
>
> Thank you for pointing out this mistake, we corrected the text.. Models only have access to vision on benchmarks that require it, ie only VisualWebArena. On the other benchmarks, those models are run with text only out of fairness.
>
> >Section 6.2: The last paragraph states that: "This result suggests that the BrowserGym API for browser interaction is more adapted for action-oriented tasks than for pure question-answering tasks.", but it is not clear what problem/limitation is causing the low performance and thus this conclusion.
>
> We updated this part of the text to describe more clearly our interpretation of those results.
>
> ## Minor/Non-essential changes
>
> We thank reviewer dn9K who provided a very thorough and valuable list of suggested changes. We discuss below the non-trivial changes that entail a response, otherwise all suggested changes not discussed here have been fixed in the updated version of the manuscript. Thank you again for these suggestions which contribute to improving the quality of our paper.
>
> >Section 3.1 - Extra element properties: Define and explain the SoM indicator and its purpose.
>
> We clarified this in the text, which now discusses the concept of Set-of-Marks prompting when we describe SoM indicators.
>
> >Section 5.4.2 - Reproduced results in the leaderboard: Clarify what qualifies an agent as "reproducible" and better explain the leaderboard's role (up to this point is only barely mentioned in the contributions of Section 1 therefore it should be explained what is it).
>
> The text was indeed confusing. We changed from reproducible ‘agent’ to reproducible ‘experiment’ and the text, and detailed the leaderboard’s role as demanded.
>
> >Section 5: Consider moving agent creation description to beginning of section.
>
> We respectfully disagree on this suggestion, as we believe launching experiments is the primary use case of AgentLab users, and this order flows more naturally. For new users, it’s an easier starting point to try an existing agent implementation and play with its configurations or its backbone LLMs. Creating a new agent is a more advanced feature which we do not expect all AgentLab users to use. Hence, we decided to leave the agent creation section in its current place.
>
> >Sec 6.1: The first paragraph describes the details of two configurations: agent and observation. From what I can conclude the two configurations are very distinct; if not, the description is misleading.
>
> This was indeed confusing in the text. We rephrased paragraph 1 of section 6.1 to clarify that there is only the agent configuration, which impacts both the agent’s reasoning and hope the observation is processed and used by the agent.

---

> > ### Author Response · Authors · 2025-02-02
> > **Reviewer-Specific Answer part 2**
> >
> > ## Broader impact
> >
> > > This work raises an interesting consideration regarding web agents' interaction with dynamic advertising content. Although the authors appropriately address the reproducibility challenges associated with such content, future ecosystem development could benefit from considering how to handle advertisement-related content in a standardized manner. Currently, network-related information (normally inspected against blacklists to identify and block ad sources) is not included in agent observations, and while this may not have a significant impact on current live benchmarks, it could become relevant as web agents become more sophisticated and spreaded. As the ecosystem aims to serve as a standard for the evaluation of web agents, a consideration of how to handle such content could help prevent potential future challenges in the interaction between web agents and dynamic advertising systems.
> >
> > This is indeed an interesting consideration, how the automation of web tasks by autonomous agents will affect the current advertising industry and practices on the web. A parallel can be made with search engines robots, which already parse the web for content indexing purposes. In a sense, autonomous web agents can be seen as an extension of such scrawling robots. We have added this discussion as a new paragraph in the broader impact section. Thank you for this suggestion.

---

> > > ### Comment · Reviewer_dn9K · 2025-02-05
> > > **Final minor remarks**
> > >
> > > I thank the authors for their detailed responses and revisions to the manuscript. The addition of cost metrics and evaluation details represent significant improvements as well as the sharing of output traces for public inspection. I also acknowledge their efforts in addressing the style and readability adjustments.
> > > - Regarding the overlap with [Drouin et al., ICML 2024], I appreciate the authors' clarification of the distinct scope of the two papers and confirmation that no text or figures were reused. The revision adequately addressed my initial concern about BrowserGym being presented as a core contribution in both papers.
> > > - While I maintain that journal publications should emphasize conceptual frameworks over implementation specifics to ensure enduring relevance and accessibility -- a role distinct from library documentation -- I recognize that this may be a matter of personal interpretation. I appreciate the authors' efforts to remove unnecessary code examples and will not insist further on this point.
> > >
> > > I have a few remaining points to address:
> > > - Section 5.2 still contains vague descriptions of computational requirements that could be improved by referring directly to the detailed appendix containing the specific hardware information.
> > > - An empty "Ressources and runtime" headline precedes Section 6.2.

---

> > > > ### Author Response · Authors · 2025-02-06
> > > > **Paper updates**
> > > >
> > > > We thank the reviewer for their constructive answer. We updated the main text to correct the issues mentioned as final remarks.
> > > > Thank you again for these detailed comments.

---

### Review · Reviewer_i4RC · 2024-12-30

**Summary Of Contributions:**

This paper introduces an ecosystem for web agents driven by Large Language Models (LLMs). The ecosystem is designed to standardize, simplify, and accelerate research and development in this domain. The claimed contributions are structured as follows:

1. BrowserGym Environment:
   A gym-like environment that generalizes interaction scenarios for web agents, providing a standardized interface for web task automation and evaluation.

2. Integration of Benchmarks:
   Unification of several existing web agent benchmarks under the BrowserGym framework, facilitating consistent evaluation, reproducibility, and cross-benchmark comparisons.

3. AgentLab Framework:
   A complementary framework to support the development, testing, and analysis of web agents. AgentLab includes tools such as AgentXRay for behavior analysis and dynamic prompting for advanced agent implementations.

4. Benchmarking Results:
   A comprehensive benchmark report comparing the performance of six LLMs/VLMs across all tasks in the BrowserGym ecosystem, with results published on the BrowserGym leaderboard.

**Audience:**

Yes

**Claims And Evidence:**

Yes

**Requested Changes:**

1. Verify or revise the claim regarding BrowserGym, ensuring it adheres to TMLR's requirements.
2. Provide deeper analysis and explanation of the benchmark scores. At a minimum, include information that helps readers understand how the web agents perform and the specific abilities demonstrated by the reported results.

--------

According to the latest revision, I have no further requests. The Claims and Evidence criterion is now met, particularly regarding the claim of BrowserGym and the improved transparency in evaluating agent capabilities.

**Strengths And Weaknesses:**

From my perspective, this paper is about publishing the BrowserGym environment and AgentLab framework, introducing their design, and providing some reports. It is clear that the topic aligns with the current hot trends in LLM/VLM-based web agent research and is worth reading for researchers in this area.

My main concern lies with the claim of introducing BrowserGym. An ICML 2024 paper, *"WorkArena: How capable are web agents at solving common knowledge work tasks?"*, also claims to introduce BrowserGym and is cited in the references. Given TMLR’s policy that submissions must be original and should not reuse text, figures, or results from prior work published in archival venues, it is unclear if the claim regarding BrowserGym in this paper meets the journal’s requirements for originality and evidence.

Regarding the benchmark results, I suggest refining Sections 4 or 6 to include explanations of the abilities reflected by the benchmark scores, rather than just reporting the numbers. It is important to provide users with a clear understanding of the levels of ability demonstrated on the unified evaluation platform, saving them the effort of interpreting the details of external benchmarks. If possible, categorizing the tasks across benchmarks and summarizing the high-level performance of web agents would provide better insights into their capabilities, rather than leaving readers to compare general scores without understanding which model excels at which specific abilities.

---

> ### Author Response · Authors · 2025-02-02
> **Reviewer-Specific Answer**
>
> ## Result section
>
> >Regarding the benchmark results, I suggest refining Sections 4 or 6 to include explanations of the abilities reflected by the benchmark scores, rather than just reporting the numbers. It is important to provide users with a clear understanding of the levels of ability demonstrated on the unified evaluation platform, saving them the effort of interpreting the details of external benchmarks. If possible, categorizing the tasks across benchmarks and summarizing the high-level performance of web agents would provide better insights into their capabilities, rather than leaving readers to compare general scores without understanding which model excels at which specific abilities.
>
> >Provide deeper analysis and explanation of the benchmark scores. At a minimum, include information that helps readers understand how the web agents perform and the specific abilities demonstrated by the reported results.
>
> WorkArena L1 and L2 come with a tentative sub-categories of tasks based on skills. But we observed that the success rate does not necessarily correlate with the defined skills in the category. In practice, there is a long tail of possible reasons for failure, e.g. the accessibility tree does not contain the required information for solving the task in that particular instance, or the content of the page is too large and the agent is confused. An in depth analysis would be indeed insightful. However the collection of experiments contains over 47000 episodes, up to 30 steps each. Analysing a single episode to find the point of failure is already tedious, even using AgentXRay. The human effort required to analyse a useful subset is comparable to the effort of labeling a dataset, using a substantial labelling budget. Instead, we are currently investing our effort in an automatic error/skill analysis tool using LLMs. However, this is a separate project and out of the scope of this work.
>
> In the meantime, to help gain insights on the experiments we made the following changes:
> - We added section 6.3, which is a qualitative analysis of typical errors encountered by our agent during the experiments.
> - We uploaded all the traces of our experiments on HuggingFace: https://huggingface.co/datasets/agentlabtraces/agentlabtraces (200GB). They can be downloaded and analysed with AgentXRay.
> - We added section H to the appendix, where we provide a detailed analysis of a task and explain how it succeeded/failed for some Claude and GPT-4o.
> - We plan on making AgentXRay available on HuggingFace as a live demo with a subset of our experiment.
>
> ## Code structures
>
> >The manuscript often emphasizes implementation-specific details and code structure at the expense of higher-level feature descriptions. While the technical details are valuable, this approach causes the focus to shift from the broader conceptual framework and core utilities to implementation details. A more balanced presentation that prioritizes functional descriptions over implementation specifics would make the text more accessible to a wider audience.
>
> >Avoid unnecessary code implementation details and function/class nomenclature as they reduce readability and can be easily avoided (especially in Sections 3.2, 5.1, 5.5, 5.6).
>
> The code examples in the text were stripped of unnecessary methods and code pieces that reduced the clarity. However, we decided to leave the  core pieces that allow highlighting the basic functionalities. We believe they are useful in showing the simplicity of our API and participate in the clarity of our contribution.

---

> > ### Author Response · Authors · 2025-02-04
> > **HuggingFace Demo**
> >
> > To further illustrate our approach, we’ve set up a Hugging Face space showcasing AgentXRay in action:
> >
> > https://huggingface.co/spaces/agentlabtraces/agentxray
> >
> > This instance runs a demonstration of our tool for trace analysis, currently operating on a subset of the Claude 3.5 Sonnet experiments on WorkArena L1. Given the constraints of Hugging Face's free tier, it is limited to 1GB of storage.
> >
> > We invite the reviewers and readers to explore the demo and see AgentXRay in action.

---

> > ### Comment · Reviewer_i4RC · 2025-02-07
> >
> > According to the latest revision, I have no further requests. The Claims and Evidence criterion is now met, particularly regarding the claim of BrowserGym and the improved transparency in evaluating agent capabilities.

---

### Review · Reviewer_EDqS · 2025-01-24

**Summary Of Contributions:**

In this paper, the authors present the BrowserGym framework, which unifies several state-of-the-art benchmarks to provide a unique platform for evaluating developed web agents. Using the proposed BrowserGym, six state-of-the-art (SOTA) language models (LLMs) and vision-language models (VLMs) have been tested, evaluating their performance on all the included web agent tasks.

**Audience:**

Yes

**Claims And Evidence:**

Yes

**Requested Changes:**

Following my suggestions, I ask the authors to clarify the issues I raised in the previous section. I believe this could be useful in improving the overall quality of the paper.

**Strengths And Weaknesses:**

STRENGTHS

The main contribution of this paper is the BrowserGym framework, which provides a unified test-bed for evaluating web agents based on LLMs/VLMs. The key point is that BrowserGym includes several state-of-the-art benchmarks designed to evaluate web agents' performance in different ways. This allows researchers developing new web agents to easily evaluate their performance on various tasks using this framework.

The idea of having a leaderboard is very interesting. It provides the community with a unique reference for web agents' performance and makes it easier to compare different web agents fairly.
Also the code will be released to give to chance to the other researchers to use this framework.


WEAKNESSES
The work appears to be an engineering effort that unifies several existing benchmarks. Additionally, it is unclear if AgentLab was developed by the authors. I ask the authors to clarify this point, as there is no evidence in the paper.

Another point to clarify is how new benchmarks can be easily included in this framework. As new well-designed benchmarks emerge, it would be useful if researchers could add them to this framework.

Finally, I ask the authors to discuss the computational time related to the experiments, including the hardware characteristics used.

---

> ### Author Response · Authors · 2025-02-02
> **Reviewer-Specific Answer**
>
> ## AgentLab ownership
>
> >The work appears to be an engineering effort that unifies several existing benchmarks. Additionally, it is unclear if AgentLab was developed by the authors. I ask the authors to clarify this point, as there is no evidence in the paper.
>
> AgentLab was developed by the authors of the paper. The extension of BrowserGym and the development of AgentLab constitute together our effort to unite the WebAgent community around fast reproducible research on the topic.
>
> ## New Benchmarks
>
> >Another point to clarify is how new benchmarks can be easily included in this framework. As new well-designed benchmarks emerge, it would be useful if researchers could add them to this framework.
>
> BrowserGym is highly extensible and adding a new benchmark has been relatively easy so far. But some of them require adjustments of the codebase, e.g. VisualWebArena required a generalization of the goal to allow images in the goal and not only text. Hence adding a benchmark usually goes beyond implementing a specific API and requires communication with authors. For this, we have GitHub issues and a Discord community that will welcome any discussion around contributions.

---

### Author Response · Authors · 2025-02-02
**General Answer**

We first would like to thank all 3 reviewers for evaluating our submission and providing valuable feedback. We will first summarize and address the common concerns shared by the reviewers, and then provide a one-by-one reply to each reviewer to address more specific concerns.

## TMLR Policy

Both reviewers i4RC and dn9K are concerned about the paper’s eligibility under the TMLR Guidelines and Editorial Policies.

Reviewer dn9K
> The manuscript raises significant concerns about its eligibility under TMLR Guidelines and Editorial Policies "TMLR only accepts original contributions that don’t reuse the authors’ own prior work. In particular, we do not accept submissions that are expanded versions of conference papers.". There appears to be substantial content overlap with an earlier work [Drouin et al., ICML 2024] which also introduces the BrowserGym ecosystem.

Reviewer i4RC
> My main concern lies with the claim of introducing BrowserGym. An ICML 2024 paper, "WorkArena: How capable are web agents at solving common knowledge work tasks?", also claims to introduce BrowserGym and is cited in the references. Given TMLR’s policy that submissions must be original and should not reuse text, figures, or results from prior work published in archival venues, it is unclear if the claim regarding BrowserGym in this paper meets the journal’s requirements for originality and evidence.

> Requested Changes: Verify or revise the claim regarding BrowserGym, ensuring it adheres to TMLR's requirements.

First, we would like to clarify that our paper **does not reuse any text, figures, or results** from [Drouin et al., ICML 2024]. Our paper is not an extended version of [Drouin et al., ICML 2024] as its content is entirely new and original. Besides, all the reported results come from new experiments we ran for this paper. As supporting evidence we performed an automated comparison of both manuscripts using GPT-4o, which did not detect any identical sentence or paragraph (https://chatgpt.com/share/678ec817-583c-8012-9cde-a828e884e44a). If a reviewer feels a specific paragraph has overlapping content with [Drouin et al., ICML 2024], we would be happy to rephrase or remove it.

Second, we agree that the BrowserGym library was already introduced in [Drouin et al., ICML 2024], however we would like to argue that the scope of the two papers is different:
1. the main focus of [Drouin et al., ICML 2024] is the WorkArena benchmark, and its coverage of BrowserGym is minimal (1 page).
2. the version of BrowserGym we present in the paper is more mature and includes several additional features not present in [Drouin et al., ICML 2024], including:
- support for vision-based task goals (Section 3.1)
- inclusion of new benchmarks: VisualWorkArena, WebLINX, AssistantBench (Section 4 §2)
- benchmark metadata (Section 4 §3)
- benchmark-specific evaluation parameters (Section 4 §4)
- automated backend preparation (Section 4 §5)
3. The main contribution of our paper is not BrowserGym per se, but rather the BrowserGym **ecosystem**, which is built on top of the BrowserGym library introduced by [Drouin et al., ICML 2024]. AgentLab and the unified collection of benchmarks (with VisualWebArena, WebLINX and AssistantBench) are part of that ecosystem.

To alleviate any ambiguity, we rephrased the contributions of the paper in the abstract and in Section 1 to better indicate that our contribution is an ecosystem built on top of BrowserGym, already introduced in [Drouin et al., IMCL 2024]. These changes are reflected in the updated version of the manuscript (modifications in blue).

---

> ### Author Response · Authors · 2025-02-02
> **General Answer part 2**
>
> ## Compute, hardware, runtime
>
> Both reviewers EDqS and dn9K have asked for more information on the hardware, compute usage and runtime.
>
> Reviewer dn9K
> >Section 5.2: "[...] the actual compute requirement of the job is relatively small and we can launch up to 20 tasks in parallel on a single laptop or even 50-100 on a server machine with many CPU cores and large memory" is too general. Replace vague descriptions of hardware and, if possible, provide some metrics to quantify computing time/resources.
>
> >Table 2: The results table includes a cost column that, from what I can conclude, refers to the monetary cost of the model's API for evaluation. This is not exactly clear, plus I don't see the value added in reporting this information (neither the currency nor the cost per token of the API is specified, so this only provides a sense of scale and not much else).
>
> >Section 6.2: Similarly to the request for Section 5.2, if possible, it would be helpful to have metrics that relate performance to time.
>
> Reviewer EDqS
> >Finally, I ask the authors to discuss the computational time related to the experiments, including the hardware characteristics used.
>
> We updated the result table's caption and columns to be clearer, and added section F and G in the appendix. Appendix F gives a more detailed description of the experiment costs per model. Appendix G adds information on the experiment runtimes for Claude 3.5 Sonnet, along with a description of the hardware used.
>
> Again, we are grateful to all three reviewers for their valuable feedback which helps us improve the quality of our paper. We hope that we have positively addressed their comments, and we remain available for any clarification if needed.
>
> Respectfully,
> The authors

---

### Decision · Action_Editor_vC8p · 2025-02-16

**Recommendation:** Accept with minor revision

**Comment:**

I recommend accepting this paper since all three reviewers lean toward acceptance, and the paper makes valuable contributions through BrowserGym - a standardized ecosystem for web agent evaluation with compelling cross-benchmark LLM comparisons - subject to incorporating the reviewers' suggested changes.

**Audience:**

yes.

**Claims And Evidence:**

yes.

---

> ### Author Response · Authors · 2025-02-25
> **Thanks**
>
> We thank the Action Editor for their decision. We submitted the camera-ready revision, aligned with the many revision suggested by the reviewers.
> We would also like to thank again the reviewers for their extensive comments, which led to significant improvements in our contribution.